# SLAMseq reveals potential transfer of RNA from liver to kidney in the mouse

Robert W. Hunter [1,2] ✉, Jialin Sun[1,3], Trecia Palmer[1], Alicja Czopek[1], Josselin Nespoux[1], Matthew A. Bailey [1], Neeraj Dhaun [1,2], Amy H. Buck [4] & James W. Dear [1]

Extracellular RNA (exRNA) mediates intercellular communication in lower animals; whether it serves a signalling function in mammals is uncertain. Reductionist experiments, in which a single RNA is over-expressed or tagged, have shown RNA transfer between tissues but may not be relevant to normal physiology. Here, we seek to determine the scale of RNA transfer between liver and kidney using metabolic RNA labelling in mice. We use 4-thiouracil to label RNA in hepatocytes and then detect labelled RNA in the kidney using SLAMseq: SH-Linked Alkylation for Metabolic RNA sequencing. We show that in the kidney, 5% of mRNA transcripts are labelled in health, increasing to 34% after acute hepatocellular injury. In the kidney, we do not detect labelled small RNA, but do find higher levels of the liver-enriched miRNA, miR-122 after liver injury. Our results show potential transfer of RNA from liver to kidney: a phenomenon that is augmented by liver injury. There were important limitations: we could not confidently identify transferred RNA transcripts at the single-gene level and we did not assess the physiological consequences of any RNA transfer.

Extracellular RNA (exRNA) is an important mediator of intercellular communication in plants and lower animals. Small, non-coding RNAs travel between cells—even between organisms—to induce wide-ranging changes in gene expression via RNA interference. This phenomenon participates in anti-viral responses and regulates various facets of development and homoeostasis[1–5].

In mammals, although exRNA is found ubiquitously in biofluids, its role in intercellular signalling is controversial and lacks quantitative evidence[6]. We do not know whether exRNA regulates physiologically important processes in higher organisms. Several features of mammalian exRNA biology are compatible with its serving an evolutionarily conserved signalling function. exRNA is protected from degradation by being packed into vesicular, lipoprotein and ribonucleoprotein carriers[7,8]. exRNA transport bears many hallmarks of a physiological signalling system: cellular RNA export is often selective and uptake by recipient cells may be selective[9,10] and regulated[11,12].

In experimental rodent models, exRNAs can induce physiologically relevant changes in recipient cells when single microRNAs (miRNAs) are over-expressed, tagged or blocked. For example, adipose-derived miRNAs regulate Fgf21 expression in hepatocytes[13] and bone-marrow-derived miR-155 regulates insulin metabolism in hepatocytes and adipocytes[14]. Similarly, there is evidence that miR-122 appears moves from the liver to the kidney in response to liver and systemic injury. In lipopolysaccharide-induced systemic inflammation, Rivkin et al. found elevated levels of mature miR-122 in the kidney, but not pre-miR-122, suggesting that it was not locally synthesised[15]. We subsequently tested the hypothesis that liver-derived miR-122 was travelling to the kidney in the context of acute paracetamol overdose. After paracetamol-induced liver injury, miR-122 levels rose within kidney tissue; this effect was abolished after liver miRNA biogenesis was attenuated by hepatocyte-specific Dicer knockdown[16]. It is important to seek a better understanding of how RNA signalling might mediate

[1]Edinburgh Kidney Research Group, Institute for Neuroscience and Cardiovascular Research, Queen's Medical Research Institute, University of Edinburgh, Edinburgh Bioquarter, Edinburgh EH16 4TJ, United Kingdom. [2]Department of Renal Medicine, Royal Infirmary of Edinburgh, Edinburgh Bioquarter, Edinburgh EH16 4SA, United Kingdom. [3]Department of Pharmacy, The Affiliated Hospital of Qingdao University, Qingdao 266003, China. [4]Institute of Immunology & Infection Research, School of Biological Sciences, University of Edinburgh, Edinburgh EH9 3FL, United Kingdom. ✉e-mail: robert.hunter@ed.ac.uk

liver-to-kidney crosstalk because there is a poorly-understood association between liver and kidney disease[17,18].

Such experiments perturbing single miRNAs provide compelling evidence that exRNA can in principle mediate intercellular signalling[13,14,19–23]. Nevertheless, they are unable to resolve whether such signalling occurs−or is relevant−in a complex, physiological setting. It is likely that any intercellular signalling would involve multiple different RNAs, rather than a single dominant miRNA. Therefore, we sought to determine the global pool of RNA molecules moving between organs in the mouse using metabolic RNA labelling. We labelled hepatocellular RNA in vivo to test the hypothesis that RNA travels from the liver to the kidney. Here, we show that RNA molecules −having been labelled in hepatocytes−are detected within kidney tissue. We determine the global pool of RNA molecules that move from liver to kidney in health and after acute hepatocellular injury.

## Results

### AAV8-TBG-Cre induces hepatocyte-specific expression of uracil phosphoribosyl transferase

We labelled RNA in hepatocytes using the modified nucleoside, 4-thiouridine. We sought to detect and sequence this labelled RNA within the kidney using SLAMseq: Thiol (SH)-Linked Alkylation for the Metabolic sequencing of RNA. To facilitate hepatocyte-specific RNA labelling, we used the adenoviral vector AAV8-TBG-Cre to induce hepatocyte-specific Cre expression in the floxed-stop-UPRT mouse (as per the protocol in Supplementary Fig. 1a). This is expected to remove a floxed stop cassette, turning on expression of recombinant HA-tagged uracil phosphoribosyl transferase (HA-UPRT). The AAV8-TBG-Cre vector, through the combined effects of viral tropism and the TBG (Thyroxine Binding Globulin) promoter, has been repeatedly shown to induce hepatocyte-specific Cre expression[16].

We used four complementary methods to verify that this induced hepatocyte-specific expression of HA-UPRT, and in particular did not cause off-target UPRT expression in the kidney. In genomic DNA (gDNA), Cre-mediated recombination at the UPRT locus was detected within liver but not kidney, spleen or heart (Fig. 1a and Supplementary Fig. 2a). Using podocin-UPRT mice as a positive control, we determined that this gDNA recombination assay was sensitive enough to detect recombination events occurring in as few as ~1:500,000 kidney cells (Supplementary Fig. 2b). In RNAseq data, TBG promoter sequence was more abundant in liver after AAV8-TBG-Cre delivery but was no different in kidney (Fig. 1b). When AAV8-TBG-Cre was injected into mTmG fluorescent reporter mice, new green fluorescence was detected in hepatocytes but not in the kidney (Fig. 1c and Supplementary Data File 1). At the protein level, HA-UPRT expression was detected in liver after AAV8-TBG-Cre injection but not in kidney (Fig. 1d; Supplementary Fig. 3).

### 4TU labels hepatocyte RNA

We conducted an initial experiment in small groups of male and female mice (Supplementary Fig. 1a and Supplementary Table 1). The purpose of this initial experiment was to verify that T > C conversions could be detected in liver RNA and to determine which control groups were necessary for the subsequent, larger, liver injury experiment. Four weeks after AAV8-TBG-Cre injection, mice were injected with 4TU. To allow sufficient time for 4TU to be incorporated into nascent liver RNA and then transferred to distant organs, we administered 4TU for 36 h prior to organ harvesting. In a biotinylation assay, we determined that our protocol led to the successful incorporation of the thiol label into liver RNA (Fig. 2a). RNA was extracted from liver and kidney and prepared for SLAMseq. Sample details for all SLAMseq data presented in this paper are included as Supplementary Data File 2; the multiQC

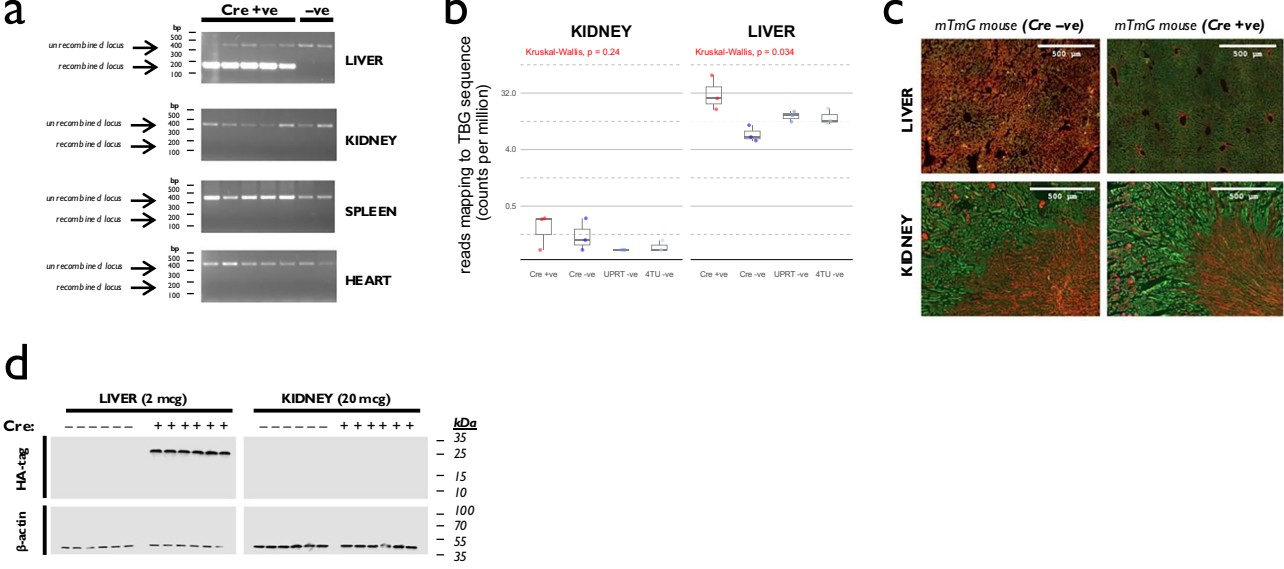

**Fig. 1 | UPRT expression is induced specifically in hepatocytes. a** Genomic DNA (gDNA) recombination assay. gDNA was extracted from liver, kidney, heart and spleen and used as templates in a PCR assay designed to detect Cre-mediated recombination at the flox-stop-UPRT locus (see Supplementary Fig. 2a for assay design). Cre-mediated recombination was detected in the liver but not in kidney, spleen or heart. Experiment repeated two times independently, with similar results. **b** TBG promoter expression. The AAV8-Cre vector contains the TBG promoter. In the RNAseq data obtained from SLAMseq experiments, these sequences were over-represented only in liver samples, following AAV8-Cre injection (p = 0.03 for comparison between Cre +ve and Cre -ve groups in liver by Kruskal-Wallis test; p = 0.24 in kidney; n = 3 mice per group). Box plot shows median (central line) and 1st and 3rd quartiles (lower and upper limits of box); whiskers define the lowest and highest values within a range extending beyond the box by 1.5x the interquartile range. **c** Fluorescent reporting of AAV8-Cre activity. The AAV8-Cre vector used to induce Cre recombination was injected into the mTmG fluorescent reporter mouse. New green fluorescence, indicative of Cre recombination, was detected only in hepatocytes and not in the kidney. (Note that there was green autofluorescence in the kidney under basal, Cre -ve conditions, particularly evident in the brush borders of the proximal tubules.) **d** Recombinant UPRT was detected by immunoblot (probing for the HA tag) in liver (2 micrograms of tissue homogenate per lane) and kidney (20 micrograms). β-actin was probed as a loading control. Cre-mediated UPRT expression was detected in liver but not kidney. Experiment repeated two times independently, with similar results. Source data are provided as a Source Data file.

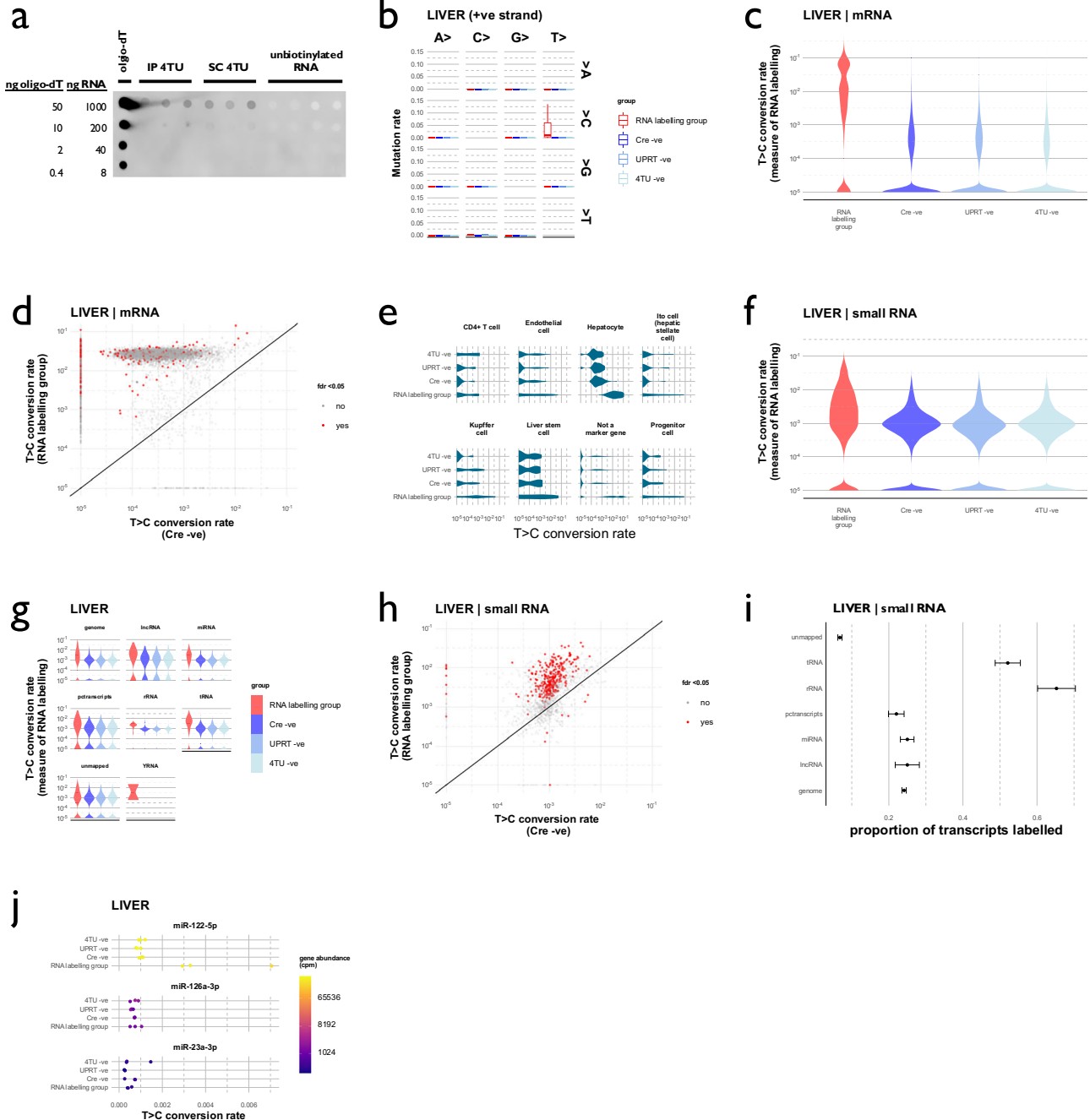

**Fig. 2 | SLAMseq labels RNA in the liver in a small initial experiment.**
**a** Biotinylation dotblot. Flox-stop-UPRT mice were treated with AAV8-TBG-Cre and then injected with 4-thiouracil as either intraperitoneal or subcutaneous injections ($n = 3$ biological replicates in each group). Incorporation of the 4TU label into liver RNA was assessed using a biotinylation assay. **b** Mutation rates in SLAMseq data. Increased rates of T > C conversion were detected on the positive strand; increased rates of A > G conversion were detected on the negative strand (Supplementary Fig. 5b). Box plot shows median (central line) and 1st and 3rd quartiles (lower and upper limits of box); whiskers define the lowest and highest values within a range extending beyond the box by 1.5x the interquartile range. **c** T > C conversion rates in mRNA SLAMseq data. Rates were higher in the RNA labelling group ($p < 2^{-16}$ for comparison with each of the negative control groups by Kruskal-Wallis test and *post hoc* Wilcoxon signed rank test). **d** Labelled mRNAs in SLAMseq data were detected by comparing gene-wise T > C conversion rates between Cre-positive (RNA

labelling) and Cre-negative (control) groups using the beta-binomial method and setting a significant false discovery rate of 0.05. Each point represents a single gene mRNA; genes for which there was a significant between-group difference in T > C conversion rate are shown in red. **e** T > C conversion rates in known hepatocyte marker genes. **f** T > C conversion rates in small RNA SLAMseq data. Rates were higher in the RNA labelling group ($p < 2^{-16}$ for comparison with each of the negative control groups by Kruskal-Wallis test and *post hoc* Wilcoxon signed rank test). **g** T > C conversion rates stratified by small RNA biotype. **h** Labelled small RNAs in SLAMseq data. **i** Labelling of small RNA in liver, stratified by RNA biotype (mean and bootstrapped 95% CI). **j** miR-122, miR-126a and miR-23a labelling. T > C conversion rates in the known hepatocyte-enriched miRNA, miR-122, and the known endothelial-enriched miRNAs, miR-126a and miR-23a. Data from male and female mice, $n = 3$ in each experimental group. Source data are provided as a Source Data file.

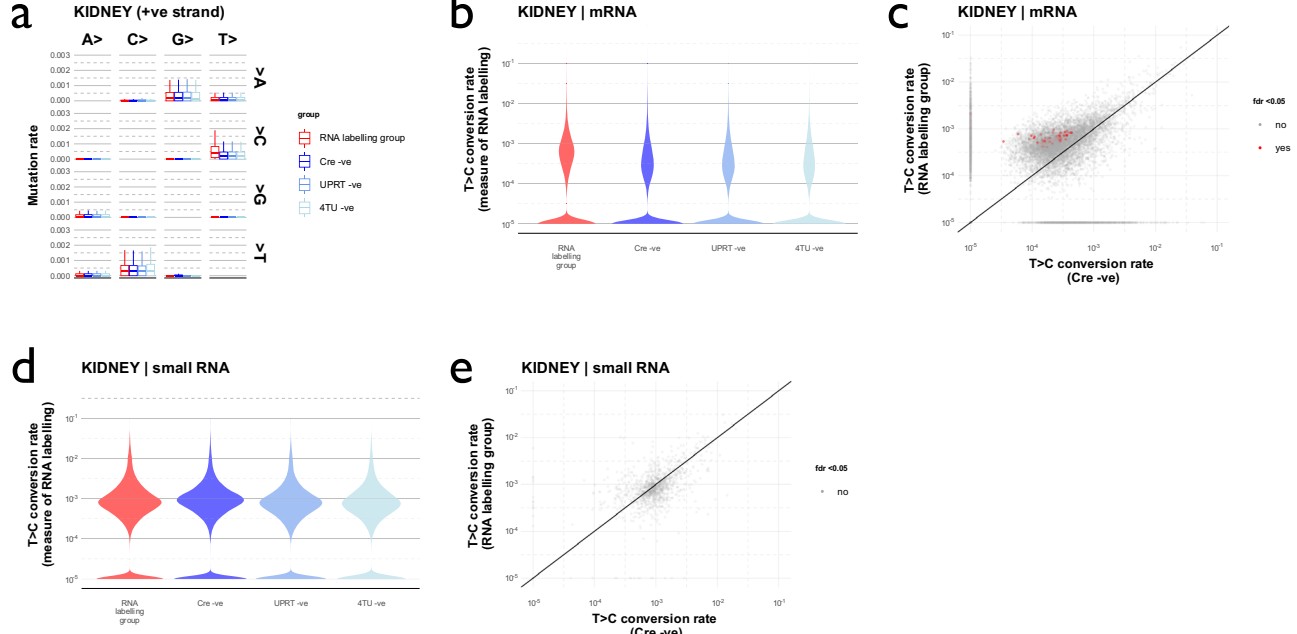

**Fig. 3 | Labelled RNA is detected within kidney in a small initial experiment.**
**a** Mutation rates in SLAMseq mRNA data. Kidney RNA was alkylated and sequenced in a SLAMseq protocol designed to sequence mRNA. Increased rates of T > C conversion were detected on the positive strand; increased rates of A > G conversion were detected on the negative strand (Supplementary Fig. 5f). Box plot shows median (central line) and 1st and 3rd quartiles (lower and upper limits of box); whiskers define the lowest and highest values within a range extending beyond the box by 1.5x the interquartile range. **b** T > C conversion rates in SLAMseq mRNA data. Rates were higher in the RNA labelling group ($p < 2^{-16}$ for comparison with each of the negative control groups by Kruskal-Wallis test and *post hoc* Wilcoxon signed

rank test). **c** Labelled mRNA transcripts were detected by comparing gene-wise T > C conversion rates between Cre-positive (RNA labelling) and Cre-negative (control) groups using the beta-binomial method and setting a significant false discovery rate of 0.05. **d** T > C conversion rates in small RNA SLAMseq data. Rates were no higher in the RNA labelling group than in the Cre -ve control group ($p = 0.51$ by Kruskal-Wallis test and *post hoc* Wilcoxon signed rank test). **e** Labelled RNAs in small RNA data. In contrast to the mRNA data, no small RNA labelling was detected within the kidney. Data derived from male and female mice, $n = 3$ in each experimental group. Source data are provided as a Source Data file.

reports are included as Supplementary data file 3. We first evaluated 4TU incorporation into liver mRNA. As expected, T > C conversions were observed in the SLAMseq dataset; rates of other single nucleotide conversion were unaffected (Fig. 2b and Supplementary Figs. 4d, e and 5a, b).

T > C conversions were observed in the RNA labelling group at a rate that exceeded the baseline rate in Cre-negative, UPRT-negative and 4TU-negative control groups (Fig. 2c; Supplementary Table 3). mRNA transcripts that exhibited significantly higher rates of T > C conversion in the RNA labelling group (compared to the Cre-negative control group) were identified using a beta-binomial test (Fig. 2d and Supplementary Data Files 4, 5). RNA labelling was apparent in transcripts known to be enriched in hepatocytes (Fig. 2e). We also used SLAMseq to assess labelling of small RNAs in the liver. An excess of T > C conversion was observed within the small RNA SLAMseq dataset (Fig. 2f–h; Supplementary Tables 3 and Supplementary Data Files 4, 5). Compared to the mRNA data, T > C conversion rates were lower in the RNA labelling group and higher in the negative control groups. In the RNA labelling group, median T > C conversion rates were 0.71% (IQR: 0.04 – 4.47) for mRNA but only 0.18% (0.06–0.57) for small RNA; in the Cre -ve control group these were 0.00% (0.00–0.03) for mRNA and 0.08 (0.03–0.14) for small RNA (Supplementary Table S3). RNA labelling was detected in reads mapping to all small RNA biotypes (Fig. 2i). RNA labelling was evident in miRNAs known to be enriched in hepatocytes, including miR-122, whereas it was not evident in the known endothelial-enriched miRNA, miR-126a (Fig. 2j).

### Labelled RNA is detected in the kidney
We next used SLAMseq to look for labelled RNA in the kidney after hepatocyte-specific RNA labelling. We detected a small excess of T > C

conversions in kidney mRNA ($p < 10^{-9}$; Fig. 3a–c; Supplementary Table 3). In a gene-wise analysis, 0.6% of all mRNAs in the kidney were identified as having been labelled (Supplementary Table 4). In contrast, there was no increase in T > C conversion rate in kidney small RNA (Fig. 3d, e).

### 4TU labels hepatocyte RNA after acute hepatocellular injury
To examine how liver-to-kidney RNA transfer might change in the context of liver injury, we induced an acute hepatocellular injury by paracetamol overdose (Supplementary Fig. 1b; Supplementary Table 2). To reduce variation in the injury response, we performed these experiments only in male mice ($n = 6$), alongside a group of male mice subjected to RNA labelling under healthy conditions (4TU +ve, AAV8-TBG-Cre +ve, paracetamol −ve; $n = 6$) and male negative control groups (4TU +ve, AAV8-TBG-Cre −ve; $n = 6$ and 4TU −ve; $n = 3$). As the DMSO vehicle ameliorated paracetamol-induced liver injury when delivered intraperitoneally, we used a subcutaneous route for 4TU in these liver injury studies. We confirmed that the subcutaneous route induced RNA labelling to the same extent as intraperitoneal 4TU (Fig. 2a). In these larger experimental groups, we first characterised T > C conversions within mRNA from healthy liver to better understand how the 4TU protocol induced RNA labelling. T > C conversion rates were higher in more abundant transcripts and those with higher predicted transcription rates; they were lower in transcripts with longer predicted half-lives (Supplementary Fig. 6). To provide confidence in our primary method for quantifying T > C conversions (SLAMDUNK analysis), we re-analysed our RNAseq data using the GRANDSLAM pipeline, an alternative method of quantifying T > C conversion rates. There was a strong correlation between the RNA labelling metrics produced by both methods (Supplementary Fig. 7). In the "Labelled in

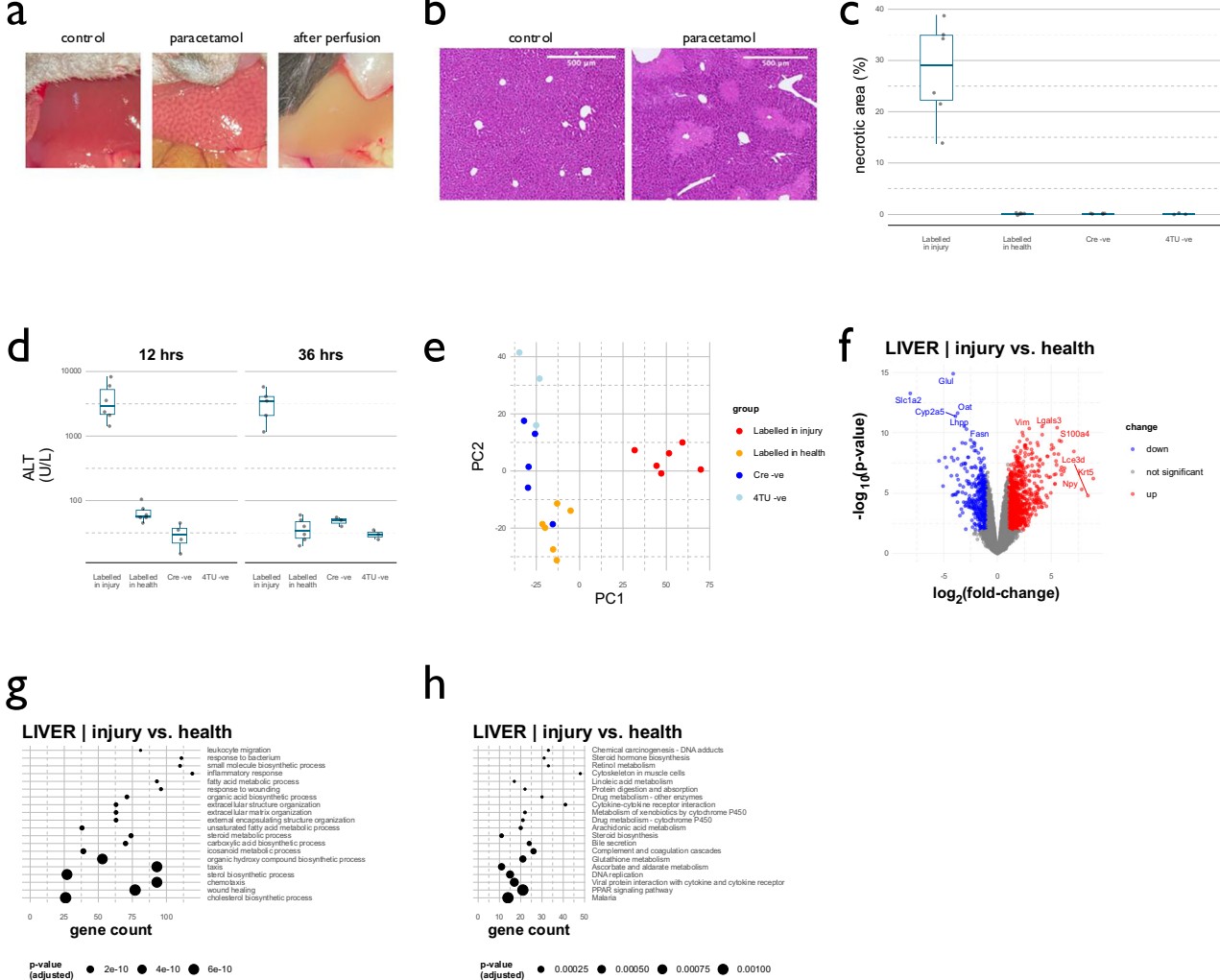

**Fig. 4 | Paracetamol overdose induces acute hepatocellular injury.**
**a** Macroscopic pathology. Representative photographs taken immediately before or after systemic perfusion. Necrosis was evident under the low-power dissecting microscope. **b** Liver histology. H&E stain of fixed liver tissue showing necrosis after paracetamol administration. **c** Quantification of the necrotic area observed on H&E sections. **d** Plasma ALT concentration at 12- and 36 h post-paracetamol. Box plot shows median (central line) and 1st and 3rd quartiles (lower and upper limits of box); whiskers define the lowest and highest values within a range extending beyond the box by 1.5x the interquartile range. **e** Principal component analysis of the RNAseq data after liver injury. There were distinct global changes in the transcriptome of the liver injury group. **f** Differentially regulated genes in the liver after paracetamol injury. Differential expression was determined using a 2-group generalised linear model. Genes that were differentially regulated with a fold-change exceeding 2 and a false-discovery rate of < 0.05 are depicted in red (if upregulated in injury) or blue (if downregulated). **g** Pathways upregulated in liver after paracetamol injury, determined by gene ontology analysis. **h** Pathways upregulated in liver after paracetamol injury in a KEGG analysis. The pathway analyses in (**g**) and (**h**) were performed with the clusterProfiler R package, using enrichGO() and enrichKEGG() respectively. These implement over-representation analyses, correcting for multiple comparisons using the Benjamini−Hochberg method. Data from male mice; $n = 6$ (labelled after paracetamol), $n = 6$ (labelled in health), $n = 5$ (Cre-negative control), $n = 3$ (4TU-negative control). Source data are provided as a Source Data file.

health" group, a median 66.7% (IQR 51.5 – 80.4%) of transcripts were deemed to be labelled by GRANDSLAM, compared to 74.5% (54.8 – 85.5%) in the "Labelled in injury" group and 3.5% (3.5 – 10.9%) in the Cre-negative control group. Together, these data confirm that the 4TU protocol achieved widespread labelling of hepatocyte mRNA. The combination of AAV8-TBG-Cre and 4TU treatment induced global transcriptional changes in the liver, with upregulation of injury response pathways including p53 signalling: a known consequence of 4TU-induced cellular toxicity (Supplementary Fig. 8a–f)[24]. Therefore, the "Labelled in health" group, whilst relatively healthy compared to the "Labelled in injury group", exhibited low levels of liver injury induced by 4TU exposure. In contrast, the labelling protocol did not induce any significant global transcriptional changes within the kidney, providing further evidence against any off-target labelling of kidney

RNA with our protocol (Supplementary Fig. 8g, h). A single dose of intraperitoneal paracetamol (300 mg/kg) induced a sublethal, acute hepatocellular injury with extensive necrosis (Fig. 4a–d; Supplementary Fig. 9; Supplementary Data File 6). mRNA from liver exhibited global transcriptional changes compared to control groups (Fig. 4e, f); differentially regulated genes were enriched in GO and KEGG terms relating to inflammation and repair, as well as homeostatic hepatocyte functions such as lipid and xenobiotic metabolism (Fig. 4g, h). Mice treated with paracetamol had elevated serum creatinine concentrations after 36 h, consistent with a pre-renal injury (Supplementary Fig. 10a). Paracetamol administration did not induce any histological changes in the kidney, nor any renal expression of the injury-response molecule KIM1 (Supplementary Fig. 10b–g; Supplementary Data File 7). Transcriptional changes within the kidney reflected differential

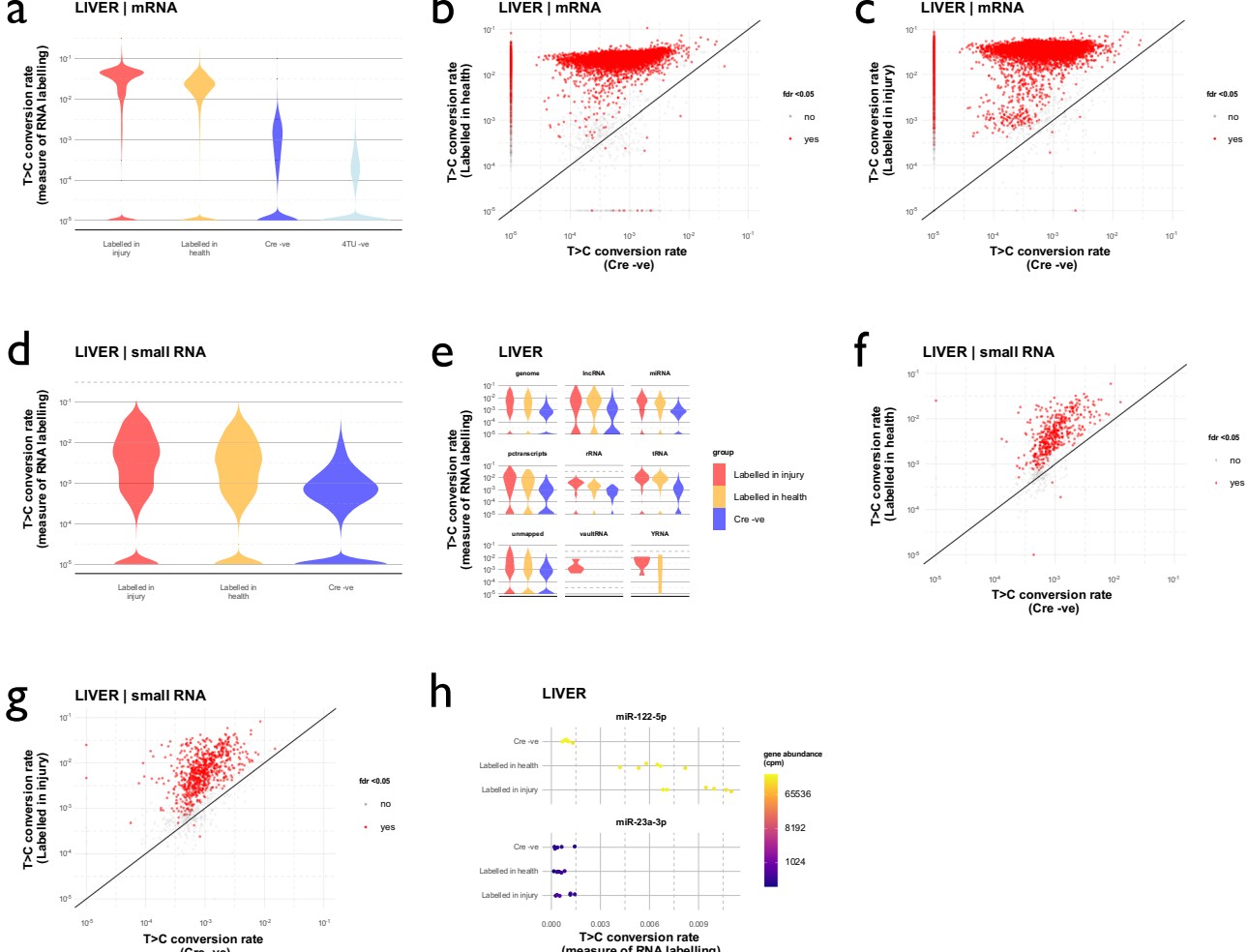

**Fig. 5 | RNA labelling in the liver after injury. a** T > C conversion rates in mRNA SLAMseq data. Rates were higher in the RNA labelling groups than the Cre -ve control group ($p < 2^{-16}$ by Kruskal-Wallis test and *post hoc* Wilcoxon signed rank tests). **b** Labelled transcripts in health. Transcripts in mRNA SLAMseq data that were deemed to be labelled by the beta-binomial test are depicted in red. **c** Labelled transcripts after injury. **d** T > C conversion rates in smallRNA. Rates were higher in the RNA labelling groups than the Cre -ve control group ($p < 2^{-16}$ by Kruskal-Wallis test and *post hoc* Wilcoxon signed rank tests). **e** T > C conversion rates in small RNA, stratified by RNA biotype. **f** Labelling of small RNA in healthy liver. **g** Labelling of small RNA in the liver after paracetamol. **h** T > C conversion rates in the known hepatocyte-enriched miRNA, miR-122, and the known endothelial-enriched miRNA, miR-23a. Data from male mice; $n = 6$ (labelled after paracetamol), $n = 6$ (labelled in health), $n = 5$ (Cre-negative control), $n = 3$ (4TU-negative control). Source data are provided as a Source Data file.

expression of metabolic and solute transport pathways after paracetamol treatment (Supplementary Fig. 10h, i).

The RNA labelling protocol achieved robust labelling of liver mRNA (Fig. 5a–c and Supplementary Fig. 11) and small RNA (Fig. 5d–h and Supplementary Fig. 12) in the context of paracetamol overdose. As in the initial experiment, T > C conversion rates differed in the small RNA and mRNA data, being lower in small RNA for the RNA labelling groups and higher in small RNA for the Cre -ve control group (Table 1). In part this may be explained by the structure of the RNAseq datasets: the number of sampled Ts per gene was higher in mRNA than in smallRNA SLAMseq data (mean 580 *vs.* 7 Ts per gene respectively).

In a gene-wise analysis, 92% of all mRNAs and 59% of small RNAs within the liver were labelled in the context of acute liver injury, compared to 89% of mRNAs and 57% of small RNAs labelled using the same labelling protocol under healthy conditions (Table 2; Supplementary Data Files 8, 9).

There was universal labelling of known hepatocyte genes under healthy conditions (Supplementary Fig. 11f, g). However, labelling was not restricted to known hepatocyte "marker" genes and labelling in non-hepatocyte marker genes was more common after liver injury

(Supplementary Fig. 11h, i). The known hepatocyte-enriched miRNA, miR-122, was labelled more strongly after paracetamol treatment, indicating increased rates of synthesis within surviving hepatocytes; there was no increase in labelling of known endothelial-enriched miRNAs (Fig. 5h).

**Labelled RNA is detected in the kidney after acute hepatocellular injury**

After liver injury, we detected increased labelled mRNA in the kidney. T > C conversion rates were significantly higher than those observed under healthy conditions (median 0.053 *vs.* 0.027% T > C conversion; $p < 10^{-12}$; Table 1; Fig. 6a–c; Supplementary Fig. 13). Labelling was observed in known hepatocyte marker genes, to a greater extent than for marker genes of other liver cell types (Fig. 6d–g). In contrast, there was no labelled small RNA in the kidney (Fig. 6g; Table 2; Supplementary Fig. S14). The liver-enriched miRNA, miR-122, did not exhibit increased rates of T > C conversion within kidney tissue (Supplementary Fig. 14f). A re-analysis of the small RNA data, using an alternative mapping strategy, gave the same result: no evidence of small RNA labelling in the kidney, despite clear labelling of small RNA in the liver

**Table 1 | T > C conversion rates in liver injury experiment**

| group | T > C conversion % (median, IQR) | | | |
|---|---|---|---|---|
| | mRNA | | small RNA | |
| | liver | kidney | liver | kidney |
| 4TU-negative | 0.000 (0.000 - 0.015) * | 0.000 (0.000 - 0.023) ** | ND | ND |
| Cre-negative | 0.017 (0.000 - 0.116) | 0.000 (0.000 - 0.036) | 0.056 (0.013 - 0.120) | 0.064 (0.019 - 0.118) |
| labelled in health | 1.861 (0.595 - 2.821) ** | 0.027 (0.000 - 0.065) ** | 0.210 (0.049 - 0.705) | 0.051 (0.000 - 0.099) ** |
| labelled in injury | 3.304 (0.938 - 4.630) ** | 0.053 (0.000 - 0.105) ** | 0.300 (0.062 - 0.990) | 0.046 (0.000 - 0.093) ** |
| 4sU | 0.000 (0.000 - 0.036) ** | 0.003 (0.000 - 0.034) ** | ND | ND |

Rates are expressed as % - i.e. a rate of 1% means conversion of 1 in 100 Ts. ND = not determined. *$p$ = 0.05; **$p < 10^{-12}$ for comparison to the Cre-negative group within that tissue by two-sided Wilcoxon signed rank test after Kruskal-Wallis rank sum test. Source data are provided as a Source Data file.

**Table 2 | RNA labelling rates in the liver injury experiment**

| tissue | group | mRNA | | | small RNA | | |
|---|---|---|---|---|---|---|---|
| | | not labelled | labelled | % | not labelled | labelled | % |
| liver | labelled in health | 1365 | 10,555 | 88.6 | 353 | 462 | 56.7 |
| liver | labelled in injury | 976 | 11,205 | 92.0 | 520 | 753 | 59.2 |
| liver | 4sU | 11,246 | 1059 | 8.6 | ND | ND | ND |
| kidney | labelled in health | 12,701 | 618 | 4.6 | 1544 | 0 | 0.0 |
| kidney | labelled in injury | 8794 | 4535 | 34.0 | 1454 | 0 | 0.0 |
| kidney | 4sU | 13,225 | 473 | 3.5 | ND | ND | ND |

The number of mRNAs and small RNAs (i.e. distinct genes) labelled in liver and kidney in different experimental groups. ND not determined. Labelling frequency of mRNA was significantly different between all experimental groups in both liver and kidney ($p < 10^{-7}$ by two-sided Chi-squared test for all between-group comparisons). Labelling frequency of small RNA was not different between experimental groups in the liver ($p = 0.28$) or the kidney (no labelling in either group). Source data are provided as a Source Data file.

(Supplementary Fig. 15). However, the liver-enriched miR-122 was more abundant in the kidney after liver injury (Fig. 6h). Of the 170 miRNAs present in the kidney RNAseq data, only miR-122 was significantly upregulated in kidney after liver injury (Fig. 6i; Supplementary Data File 10). Together, these observations suggest that at least one miRNA (and therefore perhaps other classes of small RNA) move from liver to kidney, but that our SLAMseq approach was not able to detect this.

**Characteristics of labelled mRNA in the kidney**

Our findings suggest that thiolated RNA may be transferred from liver to kidney, either as intact, potentially functional RNA molecules or as dissociated nucleotides that are subsequently incorporated into nascent mRNA in kidney cells. Although SLAMseq cannot directly distinguish between these possibilities, we attempted to infer which was more likely by examining which kidney RNAs were labelled under different conditions. We reasoned that indiscriminate labelling of kidney RNA by dissociated thiolated nucleotides would preferentially label RNA molecules transcribed at higher rates within kidney cells, hence targeting transcripts that are more abundant in kidney. Conversely, transfer of intact RNA molecules from hepatocytes would result in 4TU-dependent labelling within genes known to be more abundant within hepatocytes. However, such an analysis is confounded by a strong positive correlation between abundance in liver and kidney for most genes (Fig. 7a). We therefore included a group of mice treated with 4-thiouridine (4sU). Whereas 4TU is expected to be incorporated into nascent RNA only in UPRT-expressing hepatocytes, 4sU will be incorporated into nascent RNA in all cells (Supplementary Fig.1c). Therefore, the 4sU-treated group provided important context, enabling comparison to a dataset in which kidney transcripts are known to be labelled by dissociated nucleotides. 4sU administration induced a low level of RNA labelling, approximately equivalent to that seen in the Cre-negative, 4TU-exposed group (Supplementary Fig. 16a). The distribution of T > C conversions in the kidney differed between the 4sU- treated and 4TU-treated groups; 4sU treatment induced a smooth right-skew in

genewise delta T > C values (i.e. T > C conversion rate in 4sU-treated minus 4sU-negative groups) whereas 4TU treatment induced a bimodal distribution, in which a distinct pool of transcripts were clearly labelled (Supplementary Fig. 16b). Gene abundance within kidney and liver tissue was strongly correlated for most genes (Fig. 7a). Some genes that were detected as being labelled within the kidney were expressed at very low abundance within liver—particularly in the "Labelled in injury" group: an observation that suggests transfer of dissociated nucleotides from liver to kidney.

The probability of any kidney gene being labelled increased with gene abundance within kidney tissue. This was true for all experimental groups, but the shape of this relationship differed between groups (Fig. 7b). In genes that were present at lower abundance within kidney, the probability of being labelled by 4TU in health was greater than the probability of being labelled by 4sU, despite this dose of 4sU inducing much higher labelling rates in genes at higher abundance. This observation is compatible with the transfer of intact RNA molecules from liver to kidney, detected as labelling in a subset of low-abundance transcripts in kidney tissue. Those transcripts that were labelled and relatively abundant (cpm > 10) in uninjured liver were more likely to be labelled within the kidney in the 4TU-treated groups. They were also most likely to be labelled in the 4sU-treated group, likely reflecting the fact that these transcripts are also abundant – and hence highly transcribed—within the kidney. However, transcripts that were labelled but not abundant (cpm <10) in uninjured liver were more likely to be labelled within the kidney in 4TU-treated mice than were transcripts that were not labelled in liver; this was not the case in 4sU-treated mice (Fig. 7c). This difference between 4sU- and 4TU-treated mice is compatible with transfer of intact RNA molecules from liver to kidney in 4TU-treated mice.

Within those genes that were present in both liver and kidney RNAseq data, we attempted to test whether T > C conversion rates were disproportionately higher in those kidney transcripts that were expressed at higher levels within liver. In a differential expression

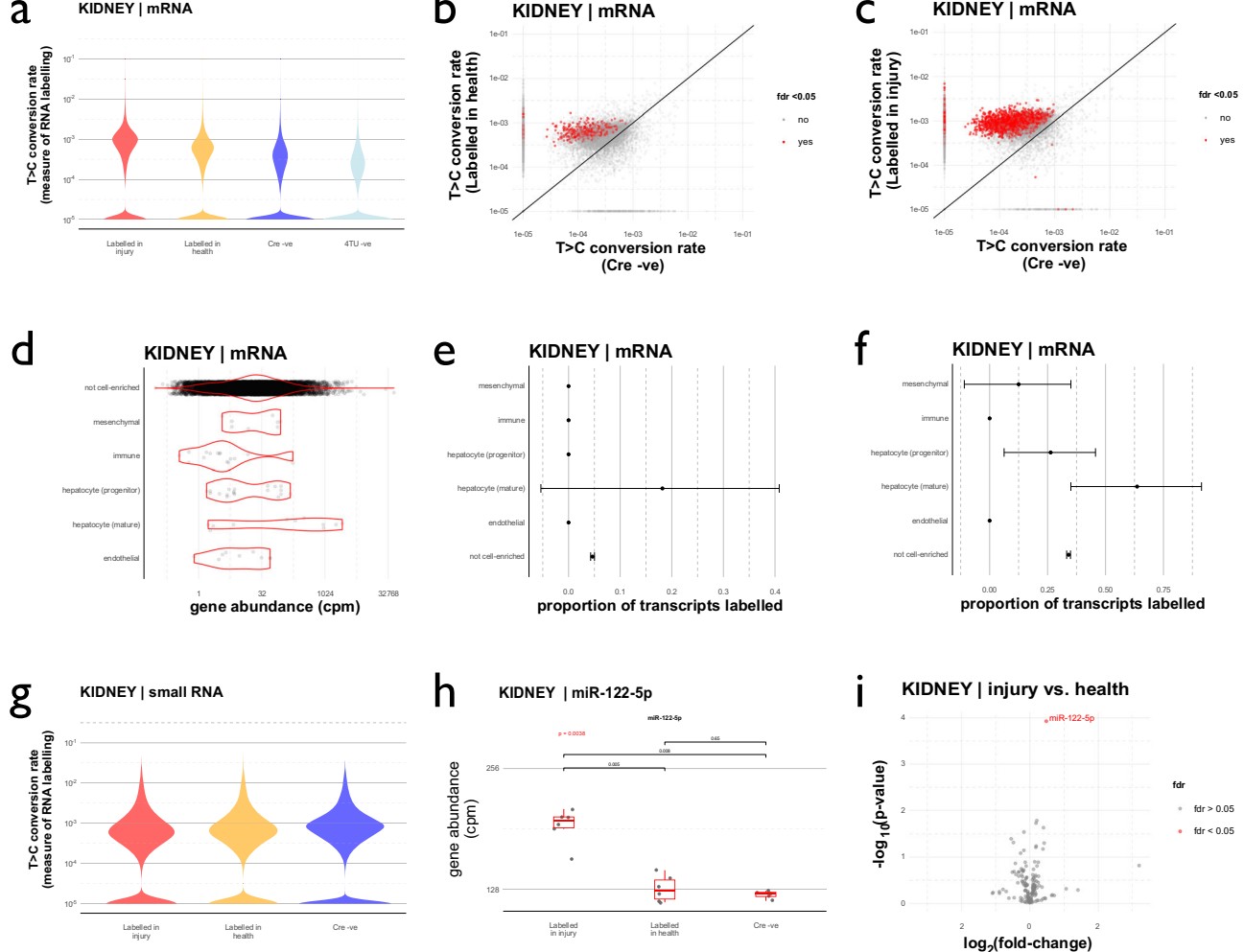

**Fig. 6 | Labelled mRNA in the kidney after liver injury. a** T > C conversion rates in kidney mRNA. Rates were higher in the RNA labelling groups than the Cre -ve control group ($p < 2^{-16}$ by Kruskal-Wallis test and *post hoc* Wilcoxon signed rank tests). **b** Labelled mRNA transcripts in kidney in health. **c** Labelled mRNA transcripts in kidney after paracetamol. **d** Abundance of liver marker genes within the kidney in health. **e** Labelling of liver marker genes within the kidney in health (mean and bootstrapped 95% CI). **f** Labelling of liver marker genes in the kidney after para-cetamol (mean and bootstrapped 95% CI). **g** T > C conversion rates in kidney small RNA. **h** miR-122 expression in kidney tissue: expression increased after liver injury ($p = 0.0038$ for comparison across all groups by Kruskal-Wallis test; *p*-values for *post-hoc* pair-wise comparisons by unpaired, two-sided Wilcoxon signed rank test

are shown on the plot). Box plot shows median (central line) and 1st and 3rd quartiles (lower and upper limits of box); whiskers define the lowest and highest values within a range extending beyond the box by 1.5x the interquartile range. **i** Differential expression analysis of all 170 miRNAs present in kidney tissue, com-paring "Labelled in health" and "Labelled in injury" groups. Differential expression was determined using a 2-group generalised linear model, adjusting for multiple comparisons using the Benjamini–Hochberg method. miR-122 was the only miRNA with significantly altered (fdr < 0.05) expression after liver injury. Data from male mice; $n = 6$ (labelled after paracetamol), $n = 6$ (labelled in health), $n = 5$ (Cre-nega-tive control), $n = 3$ (4TU-negative control). Source data are provided as a Source Data file.

analysis, we first defined genes as being liver-enriched or kidney-enriched. It would be unfair to compare crude T > C conversion rates within these gene sets within kidney tissue because the kidney-enriched genes had higher average expression within kidney tissue and would therefore be expected to have higher rates of T > C conversion. Therefore, we pulled genes at random from these liver- and kidney-enriched sets within defined cpm bins, so deriving two sets of genes that shared a similar abundance profile within kidney RNAseq data but differed in the extent to which they were expressed in liver (Fig. 7d; Supplementary Methods). Comparing T > C conversion rates, we found no consistent difference between rates of T > C conversion in liver-enriched and kidney-enriched genes within the kidney. However, when we repeated this analysis, first filtering out genes that were also labelled strongly with 4sU (and therefore likely to be labelled by transferred nucleotides), we found that liver-enriched genes exhibited higher rates of T > C conversion than did kidney-enriched genes in

RNA-labelling groups (Fig. 7e). Liver-enriched mRNAs were more strongly labelled than kidney-enriched mRNAs within the "Labelled in injury" (adjusted $p < 1^{-30}$) and "Labelled in health" (adjusted $p < 1^{-18}$) groups but not in the Cre -ve control group (adjusted $p > 0.05$). Taken together, these results are compatible with the transfer of both dis-sociated nucleotides and large mRNA fragments from liver to kidney.

**Predicted function of RNA transferred from liver to kidney**

618 transcripts (4.6%) were labelled in the kidney after treatment with 4TU under healthy conditions. This increased to 4535 transcripts (34.0%) in the context of acute hepatocellular injury. In the 4sU-treated group, 473 transcripts (3.5%) were labelled in the kidney (Table 2; Fig. 7f). Those transcripts labelled within the kidney in the 4TU- and 4sU-treated groups are listed in Supplementary Data File 8.

Using pathway analyses, we examined the known functions of mRNAs that were likely transferred from liver to kidney. We

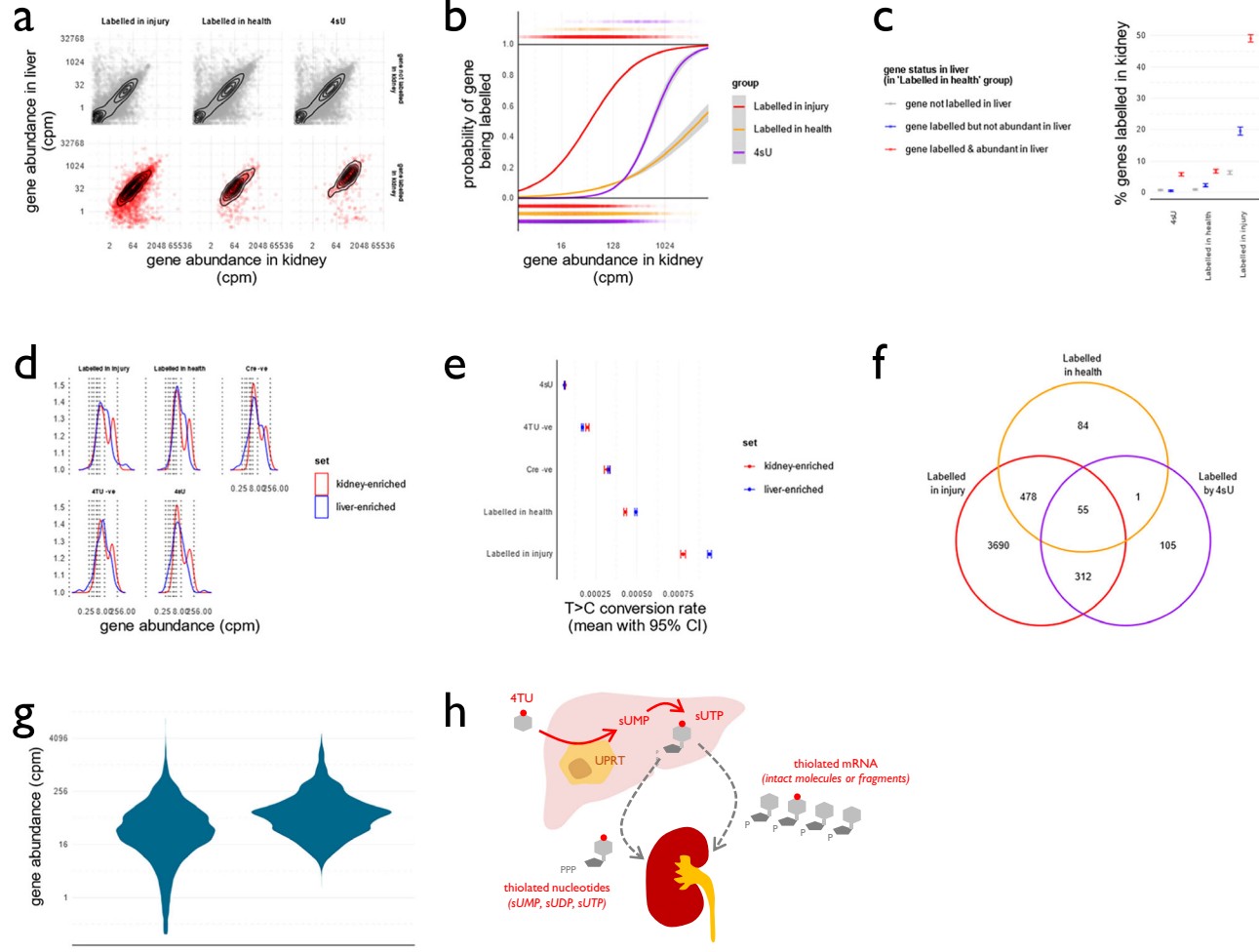

**Fig. 7 | Characterisation of labelled kidney transcripts. a** Relationship between gene abundance in liver and kidney, stratified by labelling status within kidney. **b** Probability of any given gene being labelled within the kidney mRNA SLAMseq dataset. Each point below zero represents a gene that is unlabelled; each point above 1 represents a labelled gene. The probability of any gene being labelled within each group was fitted with a sigmoidal curve. **c** Proportion of genes labelled in the kidney, stratified by gene status in liver (mean and bootstrapped 95% CI). Genes were classified according to their labelling status in the liver in the "labelled in health" group. A gene was classified as being labelled in the liver according to the beta-binomial test depicted in Fig. 5b it was deemed abundant in liver if it was detected with an adjusted cpm > 10. The proportion of genes identified as being labelled within the kidney was then determined, stratifying the results according to whether the gene was also labelled in liver. **d** Abundance of "liver-enriched" and "kidney-enriched" genes pulled at random into matching cpm bins. **e** T > C conversion rates within "liver-enriched" and "kidney-enriched" genes with matched cpm distributions. Genes exhibiting strong 4sU-dependent labelling were first filtered out. In the RNA labelling groups, rates were higher in the liver-enriched than the kidney-enriched genes ($p < 2^{-16}$ by ANOVA and two-sided *post hoc* Bonferroni correction). In the Cre -ve control group, there was no difference in the T > C conversion rate between "liver-enriched" and "kidney-enriched" genes ($p = 0.84$). **f** Number of genes labelled in kidney in different experimental groups. **g** Abundance of genes identified as being likely transferred from liver to kidney, in liver and kidney datasets. **h** Our data are compatible with the transfer of dissociated thiolated nucleotides and intact RNA molecules or large fragments of RNA molecules from liver to kidney. Data from male mice; $n = 6$ (labelled after paracetamol), $n = 6$ (labelled in health), $n = 5$ (Cre-negative control), $n = 3$ (4TU-negative control), $n = 3$ (4sU positive control). Source data are provided as a Source Data file.

reasoned that the set of 478 mRNAs that were labelled by 4TU in the kidney under both basal and injury conditions, but not by 4sU, was likely to be enriched in mRNAs that are transferred from liver to kidney. (This set is also likely to contain genes that were labelled through the transfer of dissociated nucleotides: an important limitation that we discuss below.) The vast majority of these 478 mRNAs were expressed at reasonably high levels in liver tissue (Fig. 7g). Median liver expression was 33.2 cpm; 10th centile expression 3.3 cpm, meaning that 90% of genes are expressed at >3.3 cpm in liver tissue and only 2 of the 478 genes were not present at all in the liver RNAseq data. In gene ontology and KEGG analyses, these RNAs were enriched for terms relating to RNA transcription and processing, as well as other diverse biological pathways (Supplementary Fig. 17a–c). Similarly, we reasoned that the set of 3690 mRNAs that

were labelled in kidney only in 4TU and paracetamol-treated mice, was likely to be enriched in mRNAs that are transferred from liver to kidney after liver injury (with the same caveats). Within this set, there was over-representation of mRNAs associated with catabolic and injury processes such as autophagy, mitophagy and ferroptosis (Supplementary Fig. 17d–g; Supplementary Data File 12).

## Discussion

### RNA is transferred from liver to kidney
Using SLAMseq, we have demonstrated transfer of thiolated ribonucleotides from liver to kidney in the mouse and shown that this transfer is augmented by acute hepatocellular injury. We conclude that there is transfer of both dissociated nucleotides and intact, potentially functional, RNA molecules.

Our main finding is that liver-derived RNA is apparently detectable within the kidneys. The extent to which exRNA signalling serves a physiological purpose is uncertain[6], with evidence largely derived from experiments in which single miRNAs are perturbed[13,14,19–23]. Here, we provide evidence of one necessary pre-condition of exRNA signalling: the transfer of RNA between cells. Our results suggest that any intercellular signalling would be mediated by multiple RNAs, rather than a single dominant miRNA. Whether such transfer is sufficient to induce meaningful biological effects in the recipient tissue (in this case the kidney) under physiological conditions, remains an important open question.

### A SLAMseq approach was able to track mobile extracellular RNAs

We sought to label RNA in hepatocytes with 4TU and then detect labelled RNA in liver and kidney using SLAMseq. The success of this approach is critically dependent on stringent hepatocyte-specific expression of Cre recombinase (and therefore of uracil phosphoribosyltransferase). We confirmed that there was no off-target expression of Cre and UPRT in the kidney at the level of genomic DNA, RNA and protein. We demonstrated that RNA labelling (marked by T > C conversions) was evident in liver RNA, specifically labelling known hepatocyte marker genes as well as most other (non-cell-type-restricted) transcripts. We were also able to detect T > C conversions in kidney RNA.

SLAMseq has been used previously in an attempt to tag and track small non-coding RNAs moving from the epididymis to developing sperm[25]. Although this prior attempt provided an elegant proof-of-principle that SLAMseq can be used to track exRNA moving between tissues, analysis was restricted to miRNAs in two biological replicates and lacked the resolution to identify which miRNAs were being transferred. Moreover, that study was not able to differentiate between the intercellular transfer of intact RNA molecules and dissociated thiolated nucleotides. By using a greater number of biological replicates and by comparing which set of RNAs were labelled in different states (health, liver injury, universal labelling with 4sU), we were able to identify those RNAs that were transferred from liver to kidney with moderate confidence.

### RNA is likely to be transferred intact and as dissociated nucleotides

After using 4TU to label RNA specifically in hepatocytes, we detected labelled (thiolated) RNA in the kidney. This observation might result from any of three phenomena:

i)  the off-target expression of UPRT in the kidney below our threshold for detection;
ii)  the transfer of dissociated ribonucleotides (4TUMP, 4TUDP, 4TUTP) from liver to kidney;
iii)  the transfer of intact thiolated RNA molecules.

(There could also be transfer of intact RNA that is then degraded within kidney cells, and the nucleotides re-cycled into nascent RNA molecules; our approach is unable to distinguish between this and transfer of dissociated nucleotides.)

In addition to our data showing no off-target expression of Cre and UPRT, our finding of increased rates of labelling in kidney RNA after acute hepatocellular injury is strongly suggestive of transfer of RNA from liver to kidney. Our results suggest that there is transfer of both intact RNA molecules (or large molecular fragments) and dissociated nucleotides from liver to kidney (Fig. 7h).

Compatible with dissociated nucleotide transfer, RNA labelling in the kidney was not restricted to RNAs that were labelled in liver. Indeed, there was near-universal labelling of high-abundance kidney mRNAs after liver injury, suggesting that the transfer of dissociated nucleotides exerts a strong effect on kidney RNA labelling in this context.

Several lines of evidence suggest transfer of large RNA molecules from liver to kidney. First, labelling was evident in kidney transcripts mapping to known hepatocyte "marker genes", more than for marker genes of other liver cell types. Second, a distinct population of relatively low-abundance labelled transcripts was detected in healthy mice exposed to 4TU but not in mice treated with 4sU – despite using a dose of 4sU that induced higher labelling rates in high-abundance transcripts. Third, those transcripts that were labelled but not abundant in liver were more likely to be labelled in the kidneys of 4TU-treated mice, but not so in 4sU-treated mice – despite similar labelling rates of genes that are not labelled in liver in those experimental groups. Fourth, within the kidney, labelling was stronger for liver-enriched than for kidney-enriched genes (after matching for gene abundance).

Consistent with both intact RNA and dissociated nucleotides being transferred, we observed a set of transcripts that were labelled in the kidney by both 4TU and 4sU treatments as well as distinct sets that were labelled only by 4TU.

### Labelled mRNA was detected in kidney but labelled small RNA was not

exRNA researchers have largely studied small non-coding RNAs because these are enriched in extracellular vesicles (EVs) and other exRNA carriers[26]. However, mRNAs are consistently detected within exRNA preparations and have been shown to move between cells in vivo and to exert biological effects in target cells[27–31]. Some extracellular mRNAs have been observed to enter the nucleus of recipient cells[32]. mRNAs appear to be selective packaged into EVs, in part through a mechanism dependent on their binding to the heterogenous nuclear ribonucleoprotein, HNRNPA2B1[33].

The mRNAs that we identified as being likely transferred from liver to kidney were enriched for GO terms relating to RNA transcription and processing. Intriguingly, this same enrichment for RNA-processing-related GO terms has been observed in EV-associated mRNAs[33]. Therefore, our findings extend observations that have been made by multiple investigators in diverse contexts, and suggest that exRNA signalling—if it operates in mammals—is unlikely to be restricted to small non-coding RNAs.

Perhaps surprisingly, we did not detect labelled small RNA within the kidney. This might reflect a lack of small RNA transfer from liver to kidney or that our approach was not sufficiently sensitive to detect this. The latter seems more plausible. We replicated previous work[16], showing that the hepatocyte-enriched miRNA, miR-122, became more abundant in the kidney after acute hepatocellular injury. We extended previous observations by showing in a differential expression analysis, that miR-122 was the only miRNA to exhibit significant changes in abundance in the kidney after liver injury. Given that miR-122 is highly likely to have been transferred from liver to kidney, that we did not detect increased T > C conversion in this (or any of the other small RNAs in the kidney) suggests that our SLAMseq approach was insufficiently sensitive to detect small RNA transfer.

We observed important differences between mRNA and small RNA in T > C conversion rates in both the labelling groups (relatively lower in small RNA) and negative control groups (relatively higher in small RNA). These will have necessarily reduced the sensitivity of SLAMseq to detect 4TU incorporation in small RNA. Labelling rates for small RNA were comparable to those observed elsewhere in the literature (e.g. 0.15 % was observed by Sharma et al. after labelling small RNA in epididymis)[25]. Whilst we have not rigorously explored the reasons for these differences between mRNA and small RNA, we speculate that there may be a contribution from the greater amplification required during library preparation (18 – 24 PCR cycles for small RNA vs. 13 for mRNA), which is likely to have increased the number of T > C conversions arising from PCR errors, accounting for the higher T > C conversion rates in negative control groups. This relatively lower rate of T > C conversion in the RNA labelling groups may also reflect

the increased stability of small RNAs compared to mRNA. This less extensive labelling of liver small RNA, coupled with the lower number of T residues typically present per ~22 nt miRNA than in the 3'UTR of a typical mRNA (and reflected in the big discrepancy between the number of T residues sampled per gene in our own mRNA and smallRNA data), will have impaired the sensitivity of SLAMseq in detecting labelled small RNA.

## Limitations

Our approach has several limitations. We were able to make inferences about whether a population of RNA molecules in the kidney had been labelled through the transfer of dissociated nucleotides or intact RNA from liver to kidney. However, our approach is unable to determine the route through which individual kidney transcripts are labelled. We are also not able to quantify the proportion of transcripts that were likely to have been labelled through each route. Therefore, we are not able to quantify how confidently we have identified the mRNA transcripts that move intact from liver to kidney; it is likely that there will have been contamination of these gene sets by transcripts that were labelled in the kidney through the transfer of dissociated nucleotides from the liver.

We lack a thorough understanding of some of the mechanistic aspects of RNA labelling in the context of liver injury. For example, we observed that T > C conversion rates were higher in the livers of paracetamol-treated mice than in healthy controls. This might reflect increased rates of RNA transcription in surviving hepatocytes, increased permeability of hepatocytes to 4TU or both. Our approach cannot distinguish between these, but our approach of labelling RNA in hepatocytes and then detecting this label using SLAMseq in the kidney should provide a valid method for detecting RNA transfer regardless.

Our small RNA pipeline did not account for some features that may be important when quantifying and mapping T > C conversions. For example, it did not consider RNA isoforms and untemplated additions. Future work attempting to use SLAMseq to track mobile small RNAs may need to account for these phenomena, as well as addressing the issue of poor sensitivity that we encountered.

Finally, our work does not address the physiological consequences of any RNA transfer. The extent to which this phenomenon impacts on kidney physiology remains an important, unanswered question.

## Perspectives

Our results provoke further questions and hypotheses concerning the physiological consequences of liver-to-kidney exRNA signalling. After liver injury, labelled RNA in the kidney was enriched for mRNAs participating in cellular injury pathways such as autophagy, mitophagy and ferroptosis. This is consistent with the hypothesis that hepatocyte injury provokes the release of RNAs participating in "injury-response" pathways that are then taken up by kidney cells, activating the same injury response pathways in the kidney. Such a phenomenon might be evolutionarily advantageous in the context of a pervasive acute injury (e.g. exposure to a toxin) but could be maladaptive / pro-fibrotic in response to chronic injury. It would be productive to determine to extent to which these transferred RNAs induce physiologically relevant phenotypic changes in kidney cells, and whether these could plausibly mediate the well-established association between chronic liver and chronic kidney disease[17].

## Methods

### Transgenic mice

All procedures were performed under UK Home Office licence (project licence P1D9E2AD5) in accordance with the Animals (Scientific Procedures) Act, 1986 and after review and approval by local veterinary surgeons at the University of Edinburgh. Mice had free access to water and a standard RM1 diet (Special Diet Services, Witham, UK) and were maintained at 22 – 26 °C and 50% humidity on a 12 h–12 h light–dark cycle (lights on at 7 am). In initial exploratory experiments, we included mice of both sexes. For definitive liver injury experiments, we studied only male mice because there is some sexual dimorphism in the liver injury response and we wanted to minimise biological variation when studying our primary outcome, RNA transfer.

Floxed-stop-UPRT mice[34], B6;D2-Tg(CAG-GFP,-Uprt)985Cdoe/J were purchased from Jackson laboratories and were maintained through homozygous-homozygous crosses. Mice were genotyped by Transnetyx, using PCR primers that we designed to amplify products across the 3' end of the transgene locus on chromosome 12[35]. Podocin-UPRT mice were generated by crossing homozygous floxed-stop-UPRT females and hemizygous podocin-Cre males (Tg(Nphs2-cre)1Nagy)[36].

### AAV8-TBG-Cre treatment

The AAV8.TBG.PI.Cre.rBG vector (Addgene #107787) was diluted to $6.25 \times 10^{12}$ genome copies per microlitre in sterile PBS and stored in small aliquots at −70 °C. Immediately after thawing, it was diluted 10-fold in sterile PBS and 100 mcl was injected *via* the lateral tail vein, delivering a dose of $6.25 \times 10^{10}$ genome copies per mouse. Mice were left for at least 4 weeks to allow efficient expression of recombinant UPRT before 4-thiouracil administration.

### Genomic DNA recombination assay

Tissue samples were digested with proteinase K (500 mcg/ml in 60 mM Tris, 100 mM EDTA, 0.5% SDS at 55°C for 16 hrs) and then genomic DNA was isolated by phenol-chloroform extraction and isopropanol precipitation, including an RNAse step (25 mcg/ml RNAseA for 30 min at 37°C). 40 ng (c. $1.4 \times 10^4$ genome copies) gDNA was used as a template in PCR reactions designed to amplify products from the intact or recombined floxed-stop-UPRT locus (Supplementary Fig. 2a). A common forward primer (CGTGCTGGTTATTGTGCTGT) was used with two reverse primers designed respectively to amplify a 332 bp region from the intact locus (AAGTCGTGCTGCTTCATGTG) or a 167 bp from the recombined locus (TCTCGACAAGCCCAGTTTCT). The latter primer could also amplify a 1707 bp region from the intact locus and therefore we used a relatively fast cycling programme: 24 cycles of 94 °C × 45 s, 58 °C × 15 s, 72 °C × 15 s, after a 12-cycle touch-down phase in which the annealing temperature was reduced by 1°C per cycle until landing on 58 °C.

### 4-thiouracil and 4-thiouridine treatment

4-thiouracil (4TU, Sigma 440736) was dissolved at 200 mg/ml (1560 mM) in DMSO and stored in small aliquots at −20 °C. Immediately prior to administration, this was diluted 1:10 in corn oil to give a 20 mg/ml (156 mM) solution in 90% corn oil / 10% DMSO. Mice received three 20 ml/kg body weight (=400 mg/kg = 3.12 mmoles per kg) injections either subcutaneously or intraperitoneally according to the protocols in Supplementary Fig. 1.

4-thiouridine (4sU, Sigma T4509) was dissolved at 40.6 mg/ml (156 mM) in DMSO and stored in small aliquots at −20 °C. Immediately prior to administration, this was diluted 1:10 in 0.9% NaCl to give a 15.6 mM solution in 10% DMSO. Mice received three 20 ml/kg (=81 mg/kg = 0.312 mmoles per kg) injections subcutaneously, at 33, 24 and 12 h prior to cull.

### Paracetamol-induced liver injury

Mice were fasted for 12 hrs from 9 am prior to paracetamol administration. Paracetamol (Sigma) was dissolved in hot, sterile 0.9% NaCl at 15 mg/ml and delivered by IP injection (20 ml/kg body weight = 300 mg/kg). Thereafter, mice were housed in an incubator (27 °C, 40% humidity) and provided with supplementary mashed diet, to minimise paracetamol-induced morbidity.

### Blood and tissue sampling

Blood samples (50–100 mcl) were obtained at 12 hrs after paracetamol injection by lateral tail-vein sampling. At the end of the study, mice

were euthanised by terminal perfusion under anaesthesia (pento-barbital sodium 75–150 mg per kg body weight by IP injection). Through a mid-line laparotomy, a segment of aorta was clamped inferior to the origin of the renal arteries. The aorta was then cannu-lated with "P10" BTPE tubing (outer diameter 0.61 mm), secured with a silk suture. The clamps were removed, and a terminal blood sample taken before the inferior vena cava was vented. Ice-cold PBS (30–50 ml) was perfused through the aorta until the exit fluid ran clear and all organs were visibly well-perfused (Fig. 4A).

Organs were harvested in strict order (kidneys then spleen then heart then liver), using single-use forceps and scalpels so that there was no cross-contamination from liver to other organs. Tissue samples for RNA extraction were incubated in RNALater for 18 h at 4 °C before being stored at -70 °C; tissue samples for DNA and protein extraction were snap-frozen and stored at -70 °C. Tissue samples for histology were fixed in 10% neutral buffered formalin (=4% formaldehyde) for 24 h at 4 °C before being embedded in paraffin, sectioned and stained with H&E.

Blood samples were collected into K-EDTA tubes and centrifuged for 2000 g for 10 min before storing plasma at -70 °C. Plasma alanine transaminase (ALT) activity was measured in a kinetic assay, using a commercial kit (Sentinel Diagnostics 17234H) adapted for use on a Cobas Fara analyser (Roche Diagnostics). Details on the biochemical assays for AST, total bilirubin, albumin, urea and creatinine are given in the Supplementary Methods.

### KIM-1 immunohistochemistry

Kidney injury molecule 1 (KIM1) was detected in paraffin-embedded kidney sections by immunohistochemistry using a Leica Bond-Max robot. Antigen retrieval was achieved by incubating the sections in citrate buffer (pH 6.0) for 20 min and then sections were incubated with primary antibody (goat anti-KIM1, R&D Systems AF3689 at 1:200) followed by an HRP-conjugated secondary antibody (Abcam ab97110 at 1:500) and DAB detection.

### RNA extraction and alkylation

RNA was extracted and alkylated as per published SLAMseq protocols[37,38]. Briefly, tissue samples were lysed in a monophasic phenol / guanidine isothiocyanate solution (TRIsure) and then RNA extracted by chloroform extraction and isopropanol precipitation in the presence of 20 mcg glycogen and 0.1 mM DTT. Samples were treated with DNAse (Thermo AM1907) and then cleaned on silica columns (Zymo R1013), eluting into 15 microL 1 mM DTT. RNA sam-ples were alkylated by treating with 10 mM iodoacetamide in 50 mM sodium phosphate buffer / 50% DMSO for 50 °C for 15 min. The reaction was quenched with 20 mM DTT and then RNA samples purified by ethanol precipitation. The integrity and concentration of alkylated RNA samples was assessed by automated capillary elec-trophoresis (on a LabChip GX Touch). To verify success of the alky-lation reaction, 1 mM 4-thiouracil controls were included and diluted 1:10 before being subjected to UV absorbance spectrophotometry (Nanodrop in UV-Vis mode).

### RNA sequencing and quantification of T>C conversions

Alkylated RNA samples were submitted to Lexogen who prepared and sequenced libraries. Read counts are listed in Supplementary Table 5.

For mRNAseq, 500 ng input total RNA was amplified in 13 cycles of PCR during library preparation with a QuantSeq-FWD kit v2. (QuantSeq is a method for sequencing the 3' ends of mRNA[39]). Sequencing was performed on an Illumina NextSeq 2000 sequencer, to generate single-end 100 bp reads. To map and quantify T > C conversions in the 3'UTRs of mRNA, RNAseq data were analysed using the SLAMDUNK analysis pipeline (v0.4.3)[37,38]. The command line code is given in the Supplementary Methods. The 3´end annotations used were as per the

original publication describing SLAMSeq (https://www.nature.com/articles/nmeth.4435); the Bed files are at: https://www.ncbi.nlm.nih.gov/geo/query/acc.cgi?acc=GSE99970.

For smallRNAseq, 200 ng input total RNA was amplified in 18 – 24 cycles of PCR during library preparation with a Perkin Elmer NextFlex Small RNA-Seq kit v3. Sequencing was performed on an Illumina NextSeq 2000 sequencer, to generate single-end 100 bp reads. To map and quantify T > C conversions in small RNAs, we first trimmed using cutadapt (v4.0), retaining reads with a length exceeding 15 bases. Unique reads were then grouped into families comprising one "parent" sequence and multiple "child" sequences containing one or more T > C conversions. The T > C conversion rate was computed for each parent sequence. Reads were mapped sequentially to rRNA, tRNA, miRNA, piRNA, snoRNA, snRNA, vaultRNA, YRNA, lncRNA, protein-coding RNA and whole genome reference sequences (Supplementary Table 6). Mapping was performed using Bowtie2 (v2.5.1), permitting 1 mismatch in an 18-base seed. The command line for the mapping step was:

parallel "bowtie2 -f $f -x {.} --no-unal --no-hd --reorder --sensitive -N 1 -L 18 | cut -f1,3 > $f.{/.}.hits":::: $genomes_dir/*.fa

This approach accounts for the high number of duplicated reads and unique mapping requirements of smallRNA sequencing data. Our code is publicly available (https://github.com/robertwhunter/smallSLAM; https://zenodo.org/doi/10.5281/zenodo.11083131), albeit not in the form of a robust and readily reproducible pipeline.

In an exploratory analysis, we re-analysed our data using an alternative mapping approach, in which we mapped only to the mouse genome in a single step using Bowtie (see Supplementary Methods).

### Analysis of RNA labelling and differential expression

Labelled RNAs were identified by making gene-wise, between-group comparisons of T > C conversion rates using a beta binomial test (countdata package version 1.3)[40]. To define a set of known cell-enriched (or "marker" genes) we mined the CellMarker database[41]. We selected marker genes known to be expressed in normal (i.e. non-cancer) cell types in mouse liver (Supplementary data file 11).

Differential expression analysis was performed using edgeR (v3.40.2)[42]. Low-abundance reads were filtered out and then counts were normalized using the TMM (trimmed mean of M-values) method. Differential expression was determined using a 2-group generalized linear model. Transcripts were deemed to be differentially expressed if their expression was at least 2-fold different between groups and false-discovery rate was <0.05. Gene ontology and KEGG pathway analysis was performed using the clusterProfiler package (version 4.6.2)[43].

### RNA biotinylation and dotblot

10 mcg total RNA (extracted without exposure to DTT) was bio-tinylated by mixing with 1 mcg MTSEA-biotin-XX (Biotium) in 10 mM HEPES, 1 mM EDTA, 20% DMF, pH 7.5 for 120 min at room temperature. RNA was then cleaned by chloroform extraction in phase-lock gel tubes (MaxTRACT high density tubes, Qiagen) and ethanol precipitation to remove unbound biotin. Biotinylated RNA was heated to 65 °C for 5 min and then cooled on ice, before being spotted in 2.5 mcl drops onto a dry nylon membrane (Hybond-N +, Roche 11417240001). Biotinylated oligo-dT (Pro-mega Z5261) was used as a positive control. The membrane was dried for 30 min then wet in 10x SSC (1.5 M NaCl, 150 mM tri-sodium citrate) before the RNA was cross-linked by UV light (1200 mJ cm$^{-2}$ on Spectrolinker XL-1500). Membranes were blocked for 30 min in 125 mM NaCl, 17 mM Na$_2$HPO$_4$, 7.3 mM NaH$_2$PO$_4$, 1% SDS and then incubated with Streptavidin-HRP (Amersham RPN1231, 1:2000 in blocking buffer) for 10 min. Membranes were then washed in a 1:10 dilution of blocking buffer (twice for 20 min)

then in 100 mM Tris, 100 mM NaCl, 21 mM MgCl$_2$, pH 9.5 (twice for 5 min). HRP signal was detected using ECL reagent and a LICOR Odyssey Fc imager.

## Immunoblot

Quarter kidneys, containing cortex and medulla, were homogenised in 250 mM sucrose / 20 mM triethanolamine with protease inhibitors (1% Merck Protease Inhibitor Cocktail III). The homogenate was cleared of large debris by centrifugation at 4000 $g$ for 15 min at 4 °C. Samples were prepared for SDS-PAGE by mixing with 2x tris-glycine SDS sample buffer (Invitrogen) and DTT (final concentration 50 mM) and then heating to 70 °C for 15 min. SDS-PAGE was carried out using Novex WedgeWell 4-12% Tris-glycine gels. Gels were blotted onto PVDF by wet transfer in Tris-glycine buffer with 20% methanol. Membranes were then washed in 0.2% Tween in PBS and blocked in 5% (w/v) milk powder before being incubated with primary antibody (anti-HA, CST C29F4 or anti-βactin, CST 13E5 at 1:1,200) at 4 °C overnight. After three washes in 0.2% Tween in PBS, membranes were incubated with HRP-conjugated secondary antibody (mouse anti-rabbit IgG-HRP, sc-2357 at 1:10,000) for 60 min at room temperature before being washed thrice again. HRP signal was detected using ECL reagent and a LICOR OdysseyM imager.

## Data analysis and statistics

Data were analysed in R (version 4.2.2)[44] using the Tidyverse package (version 2.0.0)[45].

To facilitate plotting of gene abundance on a log axis, we added a small "nudge" of $10^{-1}$ to all normalised counts per million (cpm), so that zero values could be plotted. Similarly, prior to plotting T > C conversion rates, we added a "nudge" of $10^{-5}$.

We compared T > C conversion rates between groups using a Kruskal-Wallis rank sum test with *post hoc* two-sided, paired Wilcoxon signed rank tests with Bonferroni correction. We compared labelling frequencies between groups using a Pearson's Chi-squared test with Yates' continuity correction.

In adjusting for multiple comparisons in differential gene expression analysis or beta-binomial testing for labelled RNA, we deemed results to be statistically significant at a Benjamini-Hochberg false-discovery rate of 0.05. When determining the proportion of labelled genes within a given set, 95% confidence intervals were derived by the bootstrap method, sampling 10,000 times with replacement. We used the R boot package (v.1.3-28) and the "normal" method for constructing confidence intervals. If there were fewer than 10 genes in the set or the labelling rate within the set was 0 or 100%, we did not attempt to derive confidence intervals.

## Supplementary methods

See supplementary methods for:
- Biochemical assays
- Command line code for SLAMDUNK analysis
- Alternative approach for small RNA mapping
- GRAND-SLAM analysis
- Estimation of mRNA half-life and synthesis rates
- Quantification of T > C conversions in liver- *vs.* kidney-enriched genes

## Reporting summary

Further information on research design is available in the Nature Portfolio Reporting Summary linked to this article.

## Data availability

The RNAseq data generated in this study have been deposited in the European Nucleotide Archive (ENA) at EMBL-EBI under accession number PRJEB75334. The associated metadata are provided as Supplemental Data File 2. All other data supporting the findings of this study are available within the paper and its Supplementary Information. Source data are provided with this paper.

## Code availability

The code used for SLAM-DUNK analysis of mRNA SLAMseq data is already in the public domain (https://t-neumann.github.io/slamdunk/). Our code for the analysis of small RNA SLAMseq data is publicly available (https://github.com/robertwhunter/smallSLAM; https://zenodo.org/doi/10.5281/zenodo.11083131).

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

## Acknowledgements

This work was supported by grants from Wellcome (209562/Z/17/Z; **RWH**), BHF (RE/18/5/34216; **RWH**), Medical Research Scotland (ECG-1781-2022; **RWH**) and the China Scholarship Fund (202309370051; **JS**). This work made use of the resources provided by the Edinburgh Compute and Data Facility (ECDF) (http://www.ecdf.ed.ac.uk/). We were assisted by the Edinburgh Bioquarter Shared University Research Facilities (SuRF) for ALT assays (Specialist Assay Service), RNA quantification (Biomolecular Core), histological processing and imaging (Histology Core). We are grateful to Richard Coward (University of Bristol) for generously providing podocin-Cre mice and to Olivia Matthews for sharing images of AAV8-TBG-Cre-injected mTmG mice. We are grateful to Nacho Vinuela-Fernandez, Kylie Matchett and Melisande Addison for advice in refining the paracetamol injury protocol. Sujai Kumar helped to design the smallSLAM pipeline for quantifying T > C conversions in small RNA SLAMseq datasets. Cei Abreu-Goodger gave advice on the bioinformatic analyses.

## Author contributions

**RWH** conducted most of the experimental work and analysis, wrote the first and final drafts of the paper and is the ultimate guarantor of data integrity. **JS, TP, AC and JN** conducted some of the experiments (gDNA recombination assay, HA-UPRT immunoblot, liver necrotic area quantification, generating positive control tissue for KIM1 immunohistochemistry). **MAB, ND** and **AB** assisted in the design of SLAMseq experiments and data interpretation; they revised the manuscript in its final stages. **JWD** had critical oversight of experimental design and data analysis and revised the manuscript at early and final stages.

## Competing interests

We have no relevant conflicts of interest to disclose.
