## [Peer Review File · Nature Communications]

SLAMseq reveals potential transfer of RNA from liver to kidney in the mouse

Corresponding Author: Dr Robert Hunter

Version 0:

Reviewer comments:

Reviewer #1

(Remarks to the Author)

The present paper aims to assess RNA transfer from the liver to the kidney in physiology and organ damage. The topic of inter-organ communication, and the relative role of RNA exchange, is a hot topic. The authors of this manuscript rigorously assess the transfer of RNA species using the detection of 4-thiouracil labeled RNA (specifically generated in the liver) within the kidney using SLAMseq: SH-Linked Alkylation for the Metabolic sequencing of RNA genetically modified.

The paper is technically well conducted, as several appropriate controls support the conclusion that liver-derived RNA is present within the kidney. The identification of the thiol-labeled RNAs showed that 34% RNAs were labeled after liver injury within the kidney, and associated with catabolic and injury processes such as autophagy, mitophagy and ferroptosis.

Unfortunately, the physiopathological role of this observation is still undetermined.

Main criticisms:

A) It cannot be ruled out that RNA molecules from injured hepatocytes are filtered or secreted into urine. This would be increased after liver injury, considering that paracetamol injury is often associated with acute kidney damage, with areas of denuded epithelium within tubules and increased permeability.

1. Kidney damage should be assessed, both functionally and histologically. In case of kidney damage, a model of liver-specific damage should be performed.
2. RNA should also be extracted from urine and analyzed
3. tubular cells should be sorted from the tissue, or isolated through culture conditions and RNA should be analyzed afterward.

B) The evaluation of the possible transcriptional relevance of the identified RNA should be demonstrated. The indicated 34% percentage of liver-derived transcripts within renal cells appears quite high.

1. What about other organs?
2. The authors should focus on identifying a panel of liver-specific RNAs, which should be shown to be effectively translated into proteins within the kidney (or isolated kidney cells).
3. The authors refer to the hypothesis that "hepatocyte injury provokes the release of RNAs participating in "injury-response" pathways that are then taken up by kidney cells, activating the same injury response pathways in the kidney". It would be important to prove this concept.

Minor:

The abstract is short and poorly informative.

Fig 1C: the quality of the image appears quite low.

(Remarks on code availability)

Reviewer #2

(Remarks to the Author)

In the manuscript entitled "SLAMseq reveals transfer of RNA from liver to kidney in the mouse," the authors have evaluated the transfer of RNA from the liver to the kidney in health and after acute hepatocellular injury. The authors have labelled the hepatocyte RNA and observed increased mRNA labelling in kidney transcripts after hepatocellular injury. The work is found original. However, the following concerns are there:

- To confirm the development of paracetamol-induced acute liver injury, only the plasma alanine transaminase (ALT) levels have been checked in the present study. Authors should also evaluate the changes in liver function markers, including aspartate aminotransferase (AST), alkaline phosphatase (ALP), and total bilirubin.
- The authors have mentioned that the mRNAs were enriched in mitophagy, autophagy or ferroptosis. A table can be provided to classify the mRNAs explicitly enriched in each mechanism.
- The authors have performed the GO analysis. However, the enriched biological process (BP), cellular components (CC), and molecular functions (MF) have not been added anywhere in the manuscript, which are the critical parameters of GO analysis. The authors should incorporate the same in the manuscript.
- The title indicates the transfer of RNA from the liver to the kidney. It can be modified for clarity by incorporating the condition where this transfer occurs, either healthy or liver injury. Similarly, the introduction can be improved for clarity and better understanding.
- The authors should provide better representative images for H&E staining.
- For Figure 1D, triplicate blots should be provided in the supplementary files.
- The authors should also elaborate on the potential implications of these findings in the discussion. Comparisons to existing studies are made, but these could be expanded to more critically evaluate how the current findings align or contrast with previous work. Discuss any discrepancies or confirmations between your findings and those previously published.
- The authors observed a small to no increase in labelled small RNA in acute hepatocellular injury compared to normal. What could be the potential reason that only the transfer of labelled mRNA from the liver to the kidney has been increased and not of small RNA? Please discuss.
- For better understanding, please define the abbreviations at their first appearance in the manuscript.
- There are a lot of grammar and sentence construction mistakes. The manuscript would benefit from proofreading to correct occasional grammatical errors and improve sentence structure for better readability.

(Remarks on code availability)

Reviewer #3

(Remarks to the Author)

Please note that a PDF file containing the following text as well as some Figures created while re-analysing parts of this dataset were attached to this Review.

In the manuscript "SLAMseq reveals transfer of RNA from liver to kidney in the mouse", Hunter et al. investigate the transfer of RNA between liver and kidney cells in mice by metabolic RNA labelling. Briefly, mice (healthy or after induced acute hepatocellular injury) are exposed to 4-TU which is converted to 4sU in hepatocytes by specific expression of HA-UPRT (cf. SLAM-ITseq, Matsushima et al., 2019). Active 4sU is incorporated randomly into nascent hepatocyte transcripts. RNA from liver and kidney is extracted and subjected to the SLAM-seq protocol. Finally, RNAs are sequenced by two protocols: 3'-end sequencing (Quantseq, targeting polyadenylated RNAs) and smallRNA sequencing. The resulting datasets are then analysed for evidence of liver-originating RNA in kidney data that could have been transferred as exRNA between tissues.

The study approach appears reasonable as (i) tracking exRNA with SLAM-seq was successfully done before (cf. e.g., Sharma et al., 2018), (ii) it was reported before that exRNA vesicles may contain mRNAs (or fragments thereof) and (iii) it was reported that, e.g., miR-122 that was secreted from (injured) liver cells entered kidney cells and reduced expression of Epo therein (cf. Rivkin et al., 2016). While the authors are targeting a relevant and significant research question, I have considerable doubts that the presented data supports the claims made in the paper.

Major comments

The main claims in this paper are based on the bioinformatics analysis of the SLAM-seq Quantseq data (I will discuss the smallRNA data below), which relies on strict hepatocyte-specific expression of HA-UPRT (as the authors acknowledge and confirm in Figure 1), but also on the accurate interpretation of T-to-C conversion data and filtering of alternative sources for such observed mismatches (background conversion rates, sequencing error rates, SNPs, misaligned reads, duplicated reads, etc.). This is particularly important in this study as the reported T-to-C conversion rates in the presumed target tissue (kidney) are very low and close to reported sequencing error rates. In the current version of the paper I am, however, missing a stringent analysis in this regard.

- The current set of Figures makes it hard to judge on the data quality as key statistics such as the mismatch-rate per read

position are missing. In Figure 2B, for example, I am missing more detailed mismatch statistics per dataset and read positions (as read ends are often enriched in unspecific mismatch rates, cf. Quantseq comments below).

- Also, the shown T-to-C box shows a low median and a large IQR. In my re-analysis of (parts of) the dataset (see below), I found out that one of the replicates has a considerably higher conversion rate than all others (see below).

- In Figure 3A, the mismatch profiles show very low T-to-C conversion rates that are similar to unspecific (sequencing error) mismatch rates (e.g., C-to-T).

- Along these lines, testing for significantly higher T-to-C conversion rates needs to be stratified/normalised by the dataset specific sequencing error rate (e.g. in Table 1). My small re-analysis of parts of this dataset showed varying levels of sequencing errors (e.g., in the unlabelled datasets, see below) but I don't see where the authors correct for those.

- I am also missing an analysis of the biological replicates, e.g., how reproducible were T-to-C conversion rates between replicates?

- Also, in Fig 5+6: the observed increase in T-to-C conversion rate in liver in the paracetamol experiment could be due to a higher 4TU concentration in the respective cells. The fact that a similar increase is observed in the kidney could indeed be explained by increased liver->kidney transfer of respective RNAs but also simply due to unspecific conversions in the kidney resulting from higher 4TU levels. Indeed, Cre -ve shows higher conversion rates than 4TU -ve. How does this fit into the author's model?

- The authors report a positive correlation between T-to-C conversion rates and gene abundance and explain this (e.g. in the main text describing Figure 2C) by: "The T>C conversion rate was higher in more abundant transcripts, consistent with more abundant RNA species being transcribed at a higher rate". Transcript abundance, however, is determined not only by transcription rate but also decay rate and other factors. Genes with low transcription rates but high stability are also highly abundant. The same inaccuracy found on page 7, line 176. In the present setting, I assume that the detection of (intact) liver RNAs in the kidney data would particularly depend on the stability of those transcripts in kidney cells, it would thus be more intuitive to instead correlate conversion rates with half-lives (instead of abundances) from published mouse datasets or meta-analyses (e.g., Agarwal&Kelley, 2022).

- One recommended quality control step would be to calculate the binomial probability for observing T-to-C mismatches in old (mainly due to sequencing/mapping errors, unfiltered SNPs, etc.) and new RNA (due to the SLAM-seq conversion) for each sample as done, for example, by GRAND-SLAM (<https://doi.org/10.1093/bioinformatics/bty256>). The authors could compare these rates between replicates and sample groups. The binomial probability (if it can be fitted for these low conversion rates) would serve as a proxy for the effective 4sU concentration in the respective cells be considerably higher in liver cells.

Is there some 'liver' signature in the data?

The authors also try to find some kind of "liver" signature in the data to strengthen the claim that the observed labelled (if indeed) reads originate from liver. They claim that "Labelling was evident in kidney transcripts mapping to known hepatocyte marker genes, to a greater extent than those mapping to markers of other cell types.". I am, however, sceptical whether selecting hepatocyte marker genes from CellMarker 2.0 to show that hepatocyte-originating RNA (fragments) are imported into kidney cells is the right approach. Marker genes are typically identified by differential gene expression (DGE) which is based on a statistically significant difference between, in this case, hepatocyte data and other tissues but does not imply that there is no kidney expression at all. Thus, if the probability to be identified as 'labelled' by the authors approach depends on a transcript's abundance then this analysis would not be insightful as it would depend on the abundance of the respective cell marker genes in the target (kidney) cell type which the authors did not report on. As a quick check, I queried the ARCHS4 (<https://maayanlab.cloud/archs4/>) mouse dataset and plotted the expression in bulk-RNAseq experiments for marker genes listed in Table S11. I included only datasets that were sequenced on a HiSeq sequencer with more than 5Mio aligned reads and excluded the ones with a 'singlecellprobability' >= 0.5. I queried the "series_id", "characteristics_ch1", "extract_protocol_ch1", "source_name_ch1", "title" metadata fields for the search terms "Hepatocyte" and "Kidney" and assigned the datasets to respective sets resulting in 4131 kidney and 1086 hepatocyte datasets. Repeating this analysis with using "Liver" instead of "Hepatocyte" resulted in 18463 datasets. I then sampled 1k random RNAseq datasets from each set and plotted the results for the 11 hepatocyte marker genes and the 12 Endothelial marker genes (for reference) in Review Attachment, Fig. 1. Indeed, expression of Endothelial marker genes was considerably lower, and this possibly explains why less 'labelled' genes were found for this gene set.

Is there evidence for whole mRNA (fragments) in the kidney data?

One maybe more convincing analysis would be to look for reads mapping to 3'-ends of genes that are not expressed in the target cell type (kidney) at all but are abundant in the originating one (liver). To demonstrate, I conducted a quick-and-dirty search for genes that are highly expressed in hepatocyte datasets (median TPM > 5k) and lowly in kidney data (median TPM < 10) and plotted the data for these 28 genes (notably, only Mup3 is shared with the CellMarker list) in Review Attachment, Fig. 2. When looking at the number of T-to-C containing reads in the labelled kidney datasets, however, I found only marginal numbers (max 3) converted reads in the labelled kidney datasets. Thus, I found no evidence that one of those transcripts was transported from liver (which of course does not exclude that others might have).

I am also sceptical wrt. the discussion of the authors for distinguishing transferred transcripts (fragments) and dissociated nucleotides (p7ff.). The authors mention 478 mRNAs that were labelled by 4TU in the kidney under both basal and injury conditions, but not by 4sU (Table S11). When looking at their ARCHS4 expression, however, I found several that are very lowly expressed in hepatocyte/liver datasets as shown in Review Attachment, Fig. 3, which gives me low confidence that the respective reads stem from hepatocyte-expressed RNAs.

Altogether, I don't see a robust way for distinguish between reads stemming from kidney and liver in this dataset besides studying the binomial probabilities for T-to-C conversions as discussed above. This binomial probability should correspond to the effective 4sU concentration in the nucleus at transcription time and should arguably be much lower in the target (kidney) cells. So, if intact labelled RNAs are imported from liver into kidney, then they would have been transcribed with much higher 4sU concentrations in place and the 'density' of T-to-C conversions would be higher than for the respective RNAs transcribed in the kidney cells themselves. I doubt, however, that such an analysis (i.e., separating those two sets of RNAs or at least showing that there is more than one such set) is possible given the very low conversion rates in the kidney data and the complexity of such an analysis.

Determining genes with significantly increased T-to-C conversions.

A central claim of the authors is that they detected relatively high percentages (5% under healthy conditions and 34% after acute hepatocellular injury) of, what they call, 'labelled' transcripts. To test for this, if I understood properly, the authors applied a two-group FDR-controlled beta-binomial test to the conversion rate output from SLAMDUNK (and from their own smallRNA pipeline discussed below) to identify genes that are "labelled" in their terminology. I have several questions in this regard:

- As far as I know, the beta-binomial test was proposed for comparing count data, so the authors should explain why this test is appropriate for their purposes and extend the respective description of how the test was conducted in the methods section. Also, the respective data tables including the output from the countdata package (p-values, and adjusted p-values as well as normalized and fold-change values) as well as the abundance (in TPM) per gene should be provided (in the current version, I just found lists of genes that passed the test).

- Additionally, the authors should confirm that this method does not report false positives when applying to sets of genes that are unlabelled in liver data (see marker gene discussion above) or by using in-silico simulation data with given (fixed, low) conversion rates/sequencing error. Did the authors apply this approach to subsets of Cre-neg replicates or compare Cre-neg and 4TU-neg data?

- Also, in the text it is mentioned that (in the initial experiment) 0.6% of all analysed genes in the (labelled) kidney QuantSeq dataset were considered as labelled. What was the abundance of those genes? Did the probability to be labelled correlate with a gene's abundance? What fraction of reads were labelled and how does this compare to the liver dataset? How did the dataset with the high conversion rate (cf. Fig2B) influence this number?

Technical description of the Quantseq analysis.

The description of the Quantseq data analysis is insufficient in my opinion and needs to be extended:

- What 3'-end annotations were used and how were they calculated?
- What version of SLAMDUNK was used? What commandline was used?
- The read mismatch rates per read position should be plotted (e.g., SLAMdunk output) and provided for all datasets.
- Did the authors observe increased mismatch rates at the read ends and consider clipping the read 5'-ends? If yes, how many bases were clipped?
- What var_fraction (given the low overall conversion probabilities) was chosen to exclude SNPs?
- SLAMDUNK provides many useful QC statistics (e.g., conversion rate per read or T>C conversions over UTR positions) that can be used to determine biases in the data. Those should be added to the supplement.
- Did the authors consider controlling duplication rates with UMIs?

Technical description of the smallRNA analysis.

The smallRNA results of this study are also of concern. The high abundance of MicroRNAs in exRNA (as reported by many studies), for example, and their increased stability when compared to mRNAs should in principle make it easier to detect them if they are indeed transported from liver to kidney cells. The authors explain not finding labelled smallRNAs in kidney data by "insufficient sensitivity" (not sure what this means) of their approach, but I don't think they have presented enough evidence to exclude technical reasons related to their analysis pipeline. I have several questions/comments in this regard:

- How was the proposed pipeline evaluated? Was synthetic (simulated) data used to validate its proper function?
- Does the pipeline consider 3'/5'-isoforms and untemplated additions (tailing) of small RNAs?
- What duplication rates were observed by the authors? Why were no UMIs used for deduplication?
- How were pre-processed reads mapped to the respective reference sequences? (bowtie2 command line)

Minor comments

- Fig 1B: add box and jitter-plots (to avoid overplotting) and test for significance.
- Page 5: reference "supplemental Tables S4 & S5" should be "supplemental Tables S5 & S6"? All table/figure references should be checked.

- Fig 2G: what are 'unmapped' small RNAs? How was the T-to-C conversion rate of those estimated?
- Fig 1: subfigure labels are wrong in the caption.
- Fig 4E: Besides clustering of the "Labelled in injury" data, the PCA also shows clear clustering of the control datasets along PC2. The authors should discuss and investigate (can these differences partly be explained due to transcriptional changes induced by the AAV8-TBG vector?).
- I am missing detailed mapping statistics for the smallRNA datasets of the main experiment (cf. Fig 2I for the initial experiment)
- Software, database and annotation versions & commandlines should be added to the Supplement.
- Table S3: how were conversion rates calculated?
- Table S12: Versions of the annotation datasets are missing.
- The sample sheet I was given during the review process should be published alongside the study data.
- Where can I see T-to-C conversion rates per dataset for the main experiment?

(Remarks on code availability)

I tried to use the small RNA pipeline myself, but it does not run out of the box as it contains many local path and infrastructure (e.g., HPC configuration) references.

After fixing those paths, I manually ran "0b_pull_smallRNA_fasta.R" to download the reference sequences, however, the script failed some datasets could not be downloaded completely. For example, Rscript ../scripts/0b_pull_smallRNA_fasta.R RNA_biotypes_source_split_1.csv resulted in the following error: 'Error: lexical error: invalid char in json text [...]', and the created miRNA.fa file contained only 97 sequences. After rerunning the command, the same error occurred but the file now contained 697 entries (that were appended to the already existing file!). After deleting the file and running the script a 3rd time, it finished w/o error and contained 1860 sequences. Some other datasets also failed with the error above or with: 'Error: parse error: premature EOF'.

At this point I stopped testing the smallRNA pipeline for now.

I strongly recommend fixing those issues and testing the pipeline thoroughly with simulated data. To increase pipeline reusability, I suggest publishing a respective Docker/apptainer image and to upload the RNA reference sequences as used in this study (except for the mouse genome sequence) to a public repository (Zenodo) to enable reproducibility of the analysis.

As a side note, I wanted to let the authors know that in a pre-experiment I did, I noticed that their smallRNA dataset ERR13245034 looks very different from all other smallRNA datasets I looked at in the QC plots. It showed elevated T/C conversion rates when compared to the other liver datasets and uniform distribution of non T>C conversions over read and the UTR positions. Can the authors explain why?

Version 1:

Reviewer comments:

Reviewer #1

(Remarks to the Author)

The authors provided a reasonable response to my comments, however, some suggested experiments that should have raised interest for the paper and better detailed the destination of the identified liver-derived RNA (RNA detection within urine and within isolated tubular cells) were not performed. This was justified by the complexity of the experiments. Low sensibility and time. The message remains interesting, but its physiological relevance is less clear.

(Remarks on code availability)

The data within code can be used by the community

Reviewer #2

(Remarks to the Author)

The authors have made the suggested changes and addressed all the comments

(Remarks on code availability)

Reviewer #3

(Remarks to the Author)

I thank the authors for addressing all my comments, for providing additional analyses and QC data and particularly for rephrasing the limitations sections. I agree that the made changes strengthen the paper.

Analyzing this dataset seems challenging and overall, my confidence in the presented data remains rather low. The revised limitations section and various adaptations in the manuscript, however, now reflect the remaining doubts about the found evidence for the passage of intact/functional mRNA molecules from liver to kidney cells in mouse. As the authors address a relevant research question, I thus believe that the created datasets, the presented results and the developed methodology

will be of interest to the community.

Major comments

A fundamental question of my previous review was whether the observed T-to-C mismatches in the kidney data originate from 4sU incorporation in liver cells or whether they are simply sequencing errors or have other causes. The now provided SLAMDUNK multiQC reports are important for judging on the overall data quality of the Quantseq datasets. The injury_expt_liver_mRNA_multiqc_report shows expected T-to-C conversion rates over the 3' ends for the labeled samples, although there seems to be an artefact at the read ends (peak in "T>C conversions over reads" plot at pos 87-88). I observed the same artefact in my small re-analysis of the data in the previous review round.

The injury_expt_kidney_mRNA_multiqc_report show the same artefact but much stronger pronounced as overall "conversion" rates are much lower here. As this peak is observed in all samples (but not in the non T-to-C conversion plots) it is likely a direct consequence of the applied sequencing/bioinformatics protocol. As it cannot be excluded that this influences overall results in the kidney data (given the low observed conversion rates), the cause of this artefact should be investigated (possibly read ends need to be trimmed before running SLAMDUNK or GRANDSLAM? The GRANDSLAM mismatch plots may also be helpful to investigate this).

The injury_expt_kidney_mRNA_multiqc_report report shows low but possibly increased (as compared to the controls) T-to-C conversion rates over the 3' ends for the labeled samples. The "conversion rates per read" plot, however, also shows, e.g., C>T conversion rates on the plus strand (likely sequencing or RT errors) that are even higher than the observed T>C conversion rates for some labelled datasets and it seems as if overall mismatch percentage in labelled samples is slightly higher than in unlabeled ones. This is relevant as all downstream analyses take only T-to-C conversions into account and don't correct for sequencing errors as the authors confirmed (except for simple per-base quality filtering in SLAMDUNK). In this regard, it was very helpful that the authors repeated their analysis with GRANDSLAM which takes T-to-C sequencing errors into account to some extent (cf. <https://github.com/erhard-lab/gedi/wiki/GRAND-SLAM#more-explanations-on-grand-slam> for a discussion). To better understand the results of this analysis, it would be helpful if the authors could provide the binomial estimates from those experiments (i.e., all *binomEstimated.tsv files possibly combined in one table) and add the overall ntr plots (*ntr.png) to the Supplement as I assume that Sup. Fig 6 was created from the ntr values per gene/utr? Could GRANDSLAM reliably estimate the conversion rates (reported in the Status column)? How do the estimated conv_new measures compare between liver and kidney cells?

Minor comments

- table "509530_1_data_set_10016079_sn2zrx" (containing study accessions and other metadata) does not contain data for s10 and s34?

- In the discussion, the authors write "Our main finding is that RNA is transferred en masse from the liver to the kidneys in mice". In light of the discussion above and the acknowledged limitations of this study I encourage the authors to rephrase this sentence accordingly.

(Remarks on code availability)

Version 2:

Reviewer comments:

Reviewer #3

(Remarks to the Author)

I thank the authors again for addressing all my comments, for providing additional analyses and QC data and for revising the main manuscript.

- Re. the observed T>C artefact: Are you sure you trimmed the right end of the reads? Anyways, I assume that you applied GRANDSLAM to the BAMs created by SLAMDUNK and would therefore of course observe the same artefact in those QC plots (Fig R2A+B) which are, however, cleaner than the slamdunk/multiqc plots in my opinion. I agree that this could possibly be a mapping artefact from NextGenMap and the authors have nicely shown that overall impact on the analysis is negligible.

- Re. GRAND-SLAM data: The binomial estimates indeed confirm a slightly higher p_new for labeled kidney data (see attached plot) and adding this table to the supplement was useful in my opinion. I agree that the fact that GRANDSLAM could not reliably estimate conversion rates in kidney data is explained by the very small number of actually detected T-to-C conversions.

- Minor comment re. suppl fig 7: There is a missing full-stop and an extra one after "g,h" in the revised figure caption.

I have no further comments or suggestions.

(Remarks on code availability)

Reviewer #1 (Remarks to the Author):

The present paper aims to assess RNA transfer from the liver to the kidney in physiology and organ damage. The topic of inter-organ communication, and the relative role of RNA exchange, is a hot topic. The authors of this manuscript rigorously assess the transfer of RNA species using the detection of 4-thiouracil labelled RNA (specifically generated in the liver) within the kidney using SLAMseq: SH-Linked Alkylation for the Metabolic sequencing of RNA genetically modified.

The paper is technically well conducted, as several appropriate controls support the conclusion that liver-derived RNA is present within the kidney. The identification of the thiol-labelled RNAs showed that 34% RNAs were labelled after liver injury within the kidney, and associated with catabolic and injury processes such as autophagy, mitophagy and ferroptosis.

Unfortunately, the physiopathological role of this observation is still undetermined.

Thank you very much for your thoughtful comments. We found them very helpful in revising our manuscript.

Main criticisms:

A) It cannot be ruled out that RNA molecules from injured hepatocytes are filtered or secreted into urine. This would be increased after liver injury, considering that paracetamol injury is often associated with acute kidney damage, with areas of denudated epithelium within tubules and increased permeability.

1. Kidney damage should be assessed, both functionally and histologically. In case of kidney damage, a model of liver-specific damage should be performed.

We used a dose of paracetamol that is not expected to induce kidney damage, but we agree that it is important to verify this. We therefore provide new data showing plasma [urea] and [creatinine], kidney histology and expression of the tubular injury marker

Kidney Injury Molecule 1, KIM1 (immunohistochemistry). As positive controls for the histology and KIM1 IHC, we included mice with tubular injury induced by ischaemia-reperfusion injury and by unilateral ureteric obstruction.

These new data show that neither the RNA labelling protocol nor paracetamol exposure induce histological tubular injury or expression of KIM1. Paracetamol treatment caused a modest and statistically insignificant rise in plasma [creatinine] at 36 hours. In light of the histological data, and our observation that mice ate and drank less after paracetamol treatment, it is likely that this represents a pre-renal injury rather than structural kidney damage. These data are presented in Supplementary Fig. 10 & Supplementary Data File 7 in the revised manuscript. We report them in the main text as follows (first paragraph, p8 of tracked-changes version):

Mice treated with paracetamol had elevated serum creatinine concentrations after 36 hours, consistent with a pre-renal injury (Supplementary Fig. 10a). Paracetamol administration did not induce any histological changes in the kidney, nor any renal expression of the injury-response molecule KIM1 (Supplementary Fig. 10b-g; Supplementary data file 7). Transcriptional changes within the kidney reflected differential expression of metabolic and solute transport pathways after paracetamol treatment (Supplementary Fig. 10h-i).

In summary, there was no evidence of structural kidney injury induced by either the RNA labelling protocol or paracetamol overdose.

2. RNA should also be extracted from urine and analyzed

This would be an interesting thing to do. We have previously shown that extracellular vesicles injected into the circulation can be detected in the urine, providing a plausible route by which circulating extracellular RNA could enter the urine (Oosthuyzen et al., JASN 2016, PMID 27020854).

We did not include urine collection as part of the present study, in part to avoid additional complexity and stress for the mice in what was already a reasonably complex experimental protocol and in part because, following paracetamol administration, the

mice became oliguric. Furthermore, whilst we have not tested this rigorously, we are almost certain that the quantity of RNA that one could extract from small urine volumes would be so low as to make SLAMseq technically very challenging if not impossible.

Whilst we agree these additional data would be interesting, they would not alter the main finding of our paper: that RNA is transferred from liver to kidney.

3. tubular cells should be sorted from the tissue, or isolated through culture conditions and RNA should be analyzed afterward.

We agree that this would be a complementary approach to validate that renal tubular cells are internalising labelled RNA. However, when we have taken a similar approach in the past, (e.g. Hunter et al, bioRxiv 2021, doi: <https://doi.org/10.1101/2021.06.15.448584>), we found that the yield of RNA obtained through this route was limiting and gave a high signal:noise ratio in SLAMseq datasets. Hence our decision to analyse whole tissue homogenates in the present study, avoiding extensive sample handling.

B) The evaluation of the possible transcriptional relevance of the identified RNA should be demonstrated. The indicated 34% percentage of liver-derived transcripts within renal cells appears quite high.

1. What about other organs?

We have previously shown that knockdown of miR-122 in hepatocytes results in reduced miR-122 expression in kidney, but not heart, lung or brain (Matthews et al., EBioMedicine 2020, PMID 33232872). This observation suggested transfer of miR-122 from liver to kidney (but not the other organs tested) and was a major impetus for the present study. We cannot exclude effects on other target organs but would argue – for the reasons that we give in response to points 2 and 3 below – that this would constitute a distinct body of work.

2. The authors should focus on identifying a panel of liver-specific RNAs, which should be shown to be effectively translated into proteins within the kidney (or isolated kidney cells).

See response immediately below.

3. The authors refer to the hypothesis that "hepatocyte injury provokes the release of RNAs participating in "injury-response" pathways that are then taken up by kidney cells, activating the same injury response pathways in the kidney". It would be important to prove this concept.

We completely agree that these are really interesting research questions. Indeed, these are questions that we hope to address in the coming years. However, they are distinct from our main focus of this paper: namely establishing the extent to which RNA moves from liver to kidney. The additional work requested to test this hypothesis properly would likely require years of research time and considerable additional funding. We would argue that this is beyond the scope of the current paper, which provide an important foundation on the transfer of RNA *in vivo* that is of broad interest.

Our observation of mass RNA transfer between organs is novel and addresses a prominent question in the field. We have therefore focussed our attention in the current paper on ensuring that our methodology is comprehensive and robust so that it will have impact, rather than exploring follow-on questions regarding the extent and functional consequences of this phenomenon.

We attempted to be appropriately circumspect in the conclusions in line with this. We realise that perhaps we should be more cautious / explicit about the limitations and have revised the final results section and discussion accordingly. We now include a dedicated section on limitations in the discussion (half-way down, p16 of the tracked-changes version).

Minor:

The abstract is short and poorly informative.

We have re-written the abstract; we hope the revised version is more informative within the 150 word limit.

Fig 1C: the quality of the image appears quite low.

We have regenerated this figure using higher-resolution images and also provide high-res images as Supplementary data file 1.

Reviewer #2 (Remarks to the Author):

In the manuscript entitled "SLAMseq reveals transfer of RNA from liver to kidney in the mouse," the authors have evaluated the transfer of RNA from the liver to the kidney in health and after acute hepatocellular injury. The authors have labelled the hepatocyte RNA and observed increased mRNA labelling in kidney transcripts after hepatocellular injury. The work is found original. However, the following concerns are there:

- To confirm the development of paracetamol-induced acute liver injury, only the plasma alanine transaminase (ALT) levels have been checked in the present study. Authors should also evaluate the changes in liver function markers, including aspartate aminotransferase (AST), alkaline phosphatase (ALP), and total bilirubin.

We now provide these new data for AST and total bilirubin (Supplementary Fig. 9). Plasma ALP was consistently below the limit of detection of our assay. The results do not alter the conclusions of the paper.

- The authors have mentioned that the mRNAs were enriched in mitophagy, autophagy or ferroptosis. A table can be provided to classify the mRNAs explicitly enriched in each mechanism.

We now provide this (Supplementary data file 12).

- The authors have performed the GO analysis. However, the enriched biological process (BP), cellular components (CC), and molecular functions (MF) have not been added anywhere in the manuscript, which are the critical parameters of GO analysis. The authors should incorporate the same in the manuscript.

We now provide this (Supplementary Fig. 17).

- The title indicates the transfer of RNA from the liver to the kidney. It can be modified for clarity by incorporating the condition where this transfer occurs, either healthy or liver injury. Similarly, the introduction can be improved for clarity and better understanding.

We have re-read and edited the whole manuscript, including the introduction, to improve clarity as well as the abstract which clarifies that transfer occurs in healthy conditions and is increased following liver injury. We have not changed the title as we prefer it to be succinct and clearly convey the main message of the paper. However, we will take editorial advice on this and would, of course, be open to changing it if a better title was preferred.

- The authors should provide better representative images for H&E staining.

We now provide these as high-resolution images (Supplementary Data Files 6 & 7).

- For Figure 1D, triplicate blots should be provided in the supplementary files.

We now provide replicate blots (Supplementary Fig. 3).

• The authors should also elaborate on the potential implications of these findings in the discussion. Comparisons to existing studies are made, but these could be expanded to more critically evaluate how the current findings align or contrast with previous work. Discuss any discrepancies or confirmations between your findings and those previously published.

This is still a very young field. To our knowledge, there has been only one previous attempt to use SLAMseq to track mobile extracellular RNA (Sharma et al., Dev Cell, 2018), which we discuss. We consider the differences between that study and our one in the discussion (p13 of the tracked-changes version, paragraph 2).

• The authors observed a small to no increase in labelled small RNA in acute hepatocellular injury compared to normal. What could be the potential reason that only the transfer of labelled mRNA from the liver to the kidney has been increased and not of small RNA? Please discuss.

This is a very good question and one we have thought long and hard about! We mention potential reasons in the discussion. We present three lines of evidence that help to understand the discrepancy between mRNA and smallRNA.

First, we present a new analysis showing that the liver-enriched miRNA, miR-122 increases in kidney after liver injury. It is the only kidney miRNA to be significantly upregulated after liver injury (revised Fig. 6h-i). This is in keeping with previous work showing that miR-122 is likely transferred from liver to kidney. This suggests that one miRNA is transferred from liver to kidney (and therefore that other small RNAs may also be transferred) but that our SLAMseq approach is not sensitive enough to detect this.

Second, we report the rates of T>C conversions within each experimental group, in the mRNA and small RNA datasets. These data were included in the original manuscript, but in the revised version we have included them in the main results text (p6 of tracked-changes version, paragraph 2 & p8, paragraph 2) and have specifically commented on how these are likely to have limited the sensitivity of the smallRNA SLAMseq approach in the discussion (copied below).

Third, we now report the number of T residues sampled per gene in the mRNA and small RNA datasets (p8 of tracked-changes version, paragraph 2).

In our revised discussion, we explain how these observations are likely to explain why we did not detect labelled small RNA in the kidney (p15 of the tracked-changes version, paragraph 3 onwards):

Perhaps surprisingly, we did not detect labelled small RNA within the kidney. This might reflect a lack of small RNA transfer from liver to kidney or that our approach was not sufficiently sensitive to detect this. The latter seems more plausible. We replicated previous work¹⁶, showing that the hepatocyte-enriched miRNA, miR-122, became more abundant in the kidney after acute hepatocellular injury. We extended previous observations by showing in a differential expression analysis, that miR-122 was the only miRNA to exhibit significant changes in abundance in the kidney after liver injury. Given that miR-122 is highly likely to have been transferred from liver to kidney, that we did not detect increased T>C conversion in this (or any of the other small RNAs in the kidney) suggests

that our SLAMseq approach was insufficiently sensitive to detect small RNA transfer.

We observed important differences between mRNA and small RNA in T>C conversion rates in both the labelling groups (relatively lower in small RNA) and negative control groups (relatively higher in small RNA). These will have necessarily reduced the sensitivity of SLAMseq to detect 4TU incorporation in small RNA. Labelling rates for small RNA were comparable to those observed elsewhere in the literature (e.g. 0.15 % was observed by Sharma et al. after labelling small RNA in epididymis).³³ Whilst we have not rigorously explored the reasons for these differences between mRNA and small RNA, we speculate that there may be a contribution from the greater amplification required during library preparation (18 – 24 PCR cycles for small RNA vs. 13 for mRNA), which is likely to have increased the number of T>C conversions arising from PCR errors, accounting for the higher T>C conversion rates in negative control groups. This relatively lower rate of T>C conversion in the RNA labelling groups may also reflect the increased stability of small RNAs compared to mRNA.^{34,35} This less extensive labelling of liver small RNA, coupled with the lower number of T residues typically present per ~22nt miRNA than in the 3'UTR of a typical mRNA (and reflected in the big discrepancy between the number of T residues sampled per gene in our own mRNA and smallRNA data), will have impaired the sensitivity of SLAMseq in detecting labelled small RNA.

- For better understanding, please define the abbreviations at their first appearance in the manuscript.

Thank you; we now do this.

- There are a lot of grammar and sentence construction mistakes. The manuscript would benefit from proofreading to correct occasional grammatical errors and improve sentence structure for better readability.

We have carefully re-read and re-written throughout to improve readability.

Reviewer #3 (Remarks to the Author):

Please note that a PDF file containing the following text as well as some Figures created while re-analysing parts of this dataset were attached to this Review.

We are very grateful for such a detailed and critical review that has helped us think further about how we interpret our data. It was valuable to have an alternative perspective on many aspects. Below we explain how we have explored, and in most cases implemented, the suggestions in the revised version.

In particular, we re-analysed our data using the alternative pipeline that you suggest (GRAND-SLAM). The results agreed with – and hence support – our original analysis.

We have acted on all your suggestions, with the result that the paper is substantially improved; thank you. The biological discovery that RNA moves from liver to kidney remains a valid conclusion after this extensive reanalysis.

In the manuscript “SLAMseq reveals transfer of RNA from liver to kidney in the mouse”, Hunter et al. investigate the transfer of RNA between liver and kidney cells in mice by metabolic RNA labelling. Briefly, mice (healthy or after induced acute hepatocellular injury) are exposed to 4-TU which is converted to 4sU in hepatocytes by specific expression of HA-UPRT (cf. SLAM-ITseq, Matsushima et al., 2019). Active 4sU is incorporated randomly into nascent hepatocyte transcripts. RNA from liver and kidney is extracted and subjected to the SLAM-seq protocol. Finally, RNAs are sequenced by two protocols: 3'-end sequencing (Quantseq, targeting polyadenylated RNAs) and smallRNA sequencing. The resulting datasets are then analysed for evidence of liver-originating RNA in kidney data that could have been transferred as exRNA between tissues.

The study approach appears reasonable as (i) tracking exRNA with SLAM-seq was successfully done before (cf. e.g., Sharma et al., 2018), (ii) it was reported before that exRNA vesicles may contain mRNAs (or fragments thereof) and (iii) it was reported that, e.g., miR-122 that was secreted from (injured) liver cells entered kidney cells and reduced expression of Epo therein (cf. Rivkin et al., 2016). While the authors are targeting a relevant and

significant research question, I have considerable doubts that the presented data supports the claims made in the paper.

Major comments

The main claims in this paper are based on the bioinformatics analysis of the SLAM-seq Quantseq data (I will discuss the smallRNA data below), which relies on strict hepatocyte-specific expression of HA-UPRT (as the authors acknowledge and confirm in Figure 1), but also on the accurate interpretation of T-to-C conversion data and filtering of alternative sources for such observed mismatches (background conversion rates, sequencing error rates, SNPs, misaligned reads, duplicated reads, etc.). This is particularly important in this study as the reported T-to-C conversion rates in the presumed target tissue (kidney) are very low and close to reported sequencing error rates. In the current version of the paper I am, however, missing a stringent analysis in this regard.

We agree that these are important considerations and we have added in the additional data that you suggest below. Our overarching approach has been to ensure that we included relevant negative control groups (e.g. the Cre-negative group). By comparing Cre-positive and Cre-negative groups, it is possible to make confident inferences about 4TU-dependent T>C conversion, because all the sources of errors that you mention (background conversion rates, sequencing errors etc.) should not differ between those groups.

- The current set of Figures makes it hard to judge on the data quality as key statistics such as the mismatch-rate per read position are missing. In Figure 2B, for example, I am missing more detailed mismatch statistics per dataset and read positions (as read ends are often enriched in unspecific mismatch rates, c.f. Quantseq comments below).

We have added data quality statistics in the form of MultiQC reports (Supplementary Data File 3). There is no structural bias in the RNAseq data that would invalidate or alter the conclusions of the study.

- Also, the shown T-to-C box shows a low median and a large IQR. In my re-analysis of (parts of) the dataset (see below), I found out that one of the replicates has a considerably higher conversion rate than all others (see below).

Thank you – this is a good point. We now report T>C conversion by library (evident in the MultiQC reports mentioned above and also presented separately for clarity: Supplementary Figs. 5 & 11 – 14. There is some variation between libraries, in line with expected biological variation between replicates. There are no outliers that might invalidate between-group comparisons.

- In Figure 3A, the mismatch profiles show very low T-to-C conversion rates that are similar to unspecific (sequencing error) mismatch rates (e.g., C-to-T).

Our comment immediately below addresses this.

- Along these lines, testing for significantly higher T-to-C conversion rates needs to be stratified/normalised by the dataset specific sequencing error rate (e.g. in Table1). My small re-analysis of parts of this dataset showed varying levels of sequencing errors (e.g., in the unlabelled datasets, see below) but I don't see where the authors correct for those.

We have replicated a widely used analysis method (SLAMDUNK pipeline) that does not include correction for per-library sequencing error rate. This pipeline does include steps aimed at mitigating the effects of sequencing errors on T>C conversion rates (e.g. there is a base-quality filter on conversion calls – see Neumann et al., BMC Bioinformatics 2019, <https://pubmed.ncbi.nlm.nih.gov/31109287/>).

It is of course true that there will be some variation between libraries in sequencing error rates. However, all the samples for each experiment were processed in the same library preparation and sequencing run and we make all comparisons to relevant negative control groups (i.e. Cre-negative and 4TU-negative). Therefore, we can make accurate inferences about RNA labelling because any between-library differences in overall sequencing error rate are small (relative to T>C conversion resulting from 4TU incorporation) and will be distributed at random between treatment groups. We

confirmed that there was no apparent large variation in sequencing error rates by plotting per library T>A and T>G conversion rates (Supplementary Fig. 4).

In the GRAND-SLAM approach (see below), T>C sequencing error rates were estimated from T>A and T>G conversions, as these were found to correlate in 4TU-negative groups (Jurges et al., *Bioinformatics* 2018, <https://pubmed.ncbi.nlm.nih.gov/29949974/>). In our dataset, correlation between T>A and T>C (or T>G and T>C) conversion rates in the 4TU negative control group were reasonably poor (R^2 in linear regression models 0.11 and 0.02 respectively; Supplementary Fig. 4). Therefore, we did not attempt to use these to predict T>C sequencing error rates.

- I am also missing an analysis of the biological replicates, e.g., how reproducible were T-to-C conversion rates between replicates?

As per comment above, we now report per-library T>C conversion rates as new supplementary data. These demonstrate that T>C conversion rates were reproducible between replicates.

- Also, in Fig 5+6: the observed increase in T-to-C conversion rate in liver in the paracetamol experiment could be due to a higher 4TU concentration in the respective cells. The fact that a similar increase is observed in the kidney could indeed be explained by increased liver>kidney transfer of respective RNAs but also simply due to unspecific conversions in the kidney resulting from higher 4TU levels. Indeed, Cre -ve shows higher conversion rates than 4TU -ve. How does this fit into the author's model?

We appreciate this point and have added text to clarify that there is increased T>C conversion in liver RNA after liver injury and one plausible mechanism is increased entry of 4TU into hepatocytes. However, our new biochemical and histological data (requested by reviewer 1) show that paracetamol does not induce structural kidney injury and it is therefore unlikely that paracetamol would significantly increase entry of 4TU into kidney cells. We are careful to remain circumspect about this in the paper. For

example, we have included a new 'Limitations' section in the revised discussion (p16 of the tracked-changes version, paragraph 2 onwards):

Our approach has several limitations. We were able to make inferences about whether a population of RNA molecules in the kidney had been labelled through the transfer of dissociated nucleotides or intact RNA from liver to kidney. However, our approach is unable to determine the route through which individual kidney transcripts are labelled. We are also not able to quantify the proportion of transcripts that were likely to have been labelled through each route. Therefore, we are not able to quantify how confidently we have identified the mRNA transcripts that move intact from liver to kidney; it is likely that there will have been contamination of these gene sets by transcripts that were labelled in the kidney through the transfer of dissociated nucleotides from the liver.

We lack a thorough understanding of some of the mechanistic aspects of RNA labelling in the context of liver injury. For example, we observed that T>C conversion rates were higher in the livers of paracetamol-treated mice than in healthy controls. This might reflect increased rates of RNA transcription in surviving hepatocytes, increased permeability of hepatocytes to 4TU or both. Our approach cannot distinguish between these, but our approach of labelling RNA in hepatocytes and then detecting this label using SLAMseq in the kidney should provide a valid method for detecting RNA transfer regardless.

Our small RNA pipeline did not account for some features that may be important when quantifying and mapping T>C conversions. For example, it did not consider RNA isoforms and untemplated additions. Future work attempting to use SLAMseq to track mobile small RNAs may need to account for these phenomena, as well as addressing the issue of poor sensitivity that we encountered.

- The authors report a positive correlation between T-to-C conversion rates and gene abundance and explain this (e.g.in the main text describing Figure 2C) by: “The T>C conversion rate was higher in more abundant transcripts, consistent with more abundant RNA species being transcribed at a higher rate”. Transcript abundance, however, is determined not only by transcription rate but also decay rate and other factors. Genes with

low transcription rates but high stability are also highly abundant. The same inaccuracy found on page 7, line 176. In the present setting, I assume that the detection of (intact) liver RNAs in the kidney data would particularly depend on the stability of those transcripts in kidney cells, it would thus be more intuitive to instead correlate conversion rates with half-lives (instead of abundances) from published mouse datasets or meta-analyses (e.g., Agarwal&Kelley, 2022).

Yes – we completely accept this criticism. We have modified the text of our paper and our approach to data presentation to avoid confusion. Whilst it is tangential to the main focus of the paper (is there transfer of RNA from liver to kidney?), it is still important to fully characterise RNA labelling in the liver and therefore we have followed your suggestion and explored the relationship between T>C conversion and transcript half-life (taken from published datasets) and synthesis rate (computed from abundance and half-life).

We present the results as new supplementary data (Supplementary Fig. S6). We found that T>C conversion rate correlated directly with mRNA synthesis rate and inversely with half-life. Our aim was to label liver RNA as extensively as possible and these data show that our protocol does achieve a high proportion of labelled liver RNA (see also GRAND-SLAM data below). (We present the methods that we used for these analyses in the Supplementary Methods: p26-27 of the Supplement with tracked-changes.)

This comment also caused us to reflect on how best to present our T>C conversion rate data. It was convenient for us to plot T>C conversion rate against gene abundance in our original manuscript, but this is perhaps not particularly helpful and could be distracting. Therefore, in the revised manuscript, we prefer to present T>C conversion rates as distributions on one dimension (e.g. as density plots or violin plots) and have amended Figs. 2, 3, 5, 6 and the relevant Supplementary Figs accordingly.

- One recommended quality control step would be to calculate the binomial probability for observing T-to-C mismatches in old (mainly due to sequencing/mapping errors, unfiltered SNPs, etc.) and new RNA (due to the SLAM-seq conversion) for each sample as

done, for example, by GRAND-SLAM (<https://doi.org/10.1093/bioinformatics/bty256>). The authors could compare these rates between replicates and sample groups. The binomial probability (if it can be fitted for these low conversion rates) would serve as a proxy for the effective 4sU concentration in the respective cells be considerably higher in liver cells.

This is a good suggestion; thank you. We have reanalysed our data with the GRAND-SLAM pipeline and present this as new supplementary data (Supplementary Fig. 7). The results were in broad agreement with our original results (from the SLAM-DUNK pipeline), giving us confidence that we have quantified T>C conversion rates accurately.

We present the methods that we used for the GRAND-SLAM analysis in the Supplementary Methods: p26 of the Supplement with tracked-changes.

Is there some 'liver' signature in the data?

The authors also try to find some kind of "liver" signature in the data to strengthen the claim that the observed labelled (if indeed) reads originate from liver. They claim that "Labelling was evident in kidney transcripts mapping to known hepatocyte marker genes, to a greater extent than those mapping to markers of other cell types.". I am, however, sceptical whether selecting hepatocyte marker genes from CellMarker 2.0 to show that hepatocyte-originating RNA (fragments) are imported into kidney cells is the right approach. Marker genes are typically identified by differential gene expression (DGE) which is based on a statistically significant difference between, in this case, hepatocyte data and other tissues but does not imply that there is no kidney expression at all. Thus, if the probability to be identified as 'labelled' by the authors approach depends on a transcript's abundance then this analysis would not be insightful as it would depend on the abundance of the respective cell marker genes in the target (kidney) cell type which the authors did not report on. As a quick check, I queried the ARCHS4 (<https://maayanlab.cloud/archs4/>) mouse dataset and plotted the expression in bulk-RNAseq experiments for marker genes listed in Table S11. I included only datasets that were sequenced on a HiSeq sequencer with more than 5Mio aligned reads and excluded the ones with a 'singlecellprobability' \geq 0.5. I queried the "series_id", "characteristics_ch1", "extract_protocol_ch1", "source_name_ch1", "title" metadata fields for the search terms "Hepatocyte" and "Kidney" and assigned the datasets

to respective sets resulting in 4131 kidney and 1086 hepatocyte datasets. Repeating this analysis with using “Liver” instead of “Hepatocyte” resulted in 18463 datasets. I then sampled 1k random RNAseq datasets from each set and plotted the results for the 11 hepatocyte marker genes and the 12 Endothelial marker genes (for reference) in Review Attachment, Fig. 1. Indeed, expression of Endothelial marker genes was considerably lower, and this possibly explains why less ‘labelled’ genes were found for this gene set.

This is another very good point. There are several alternative approaches one could take here but they are all imperfect. If RNA is indeed transferred between cells, then that challenges the very concept of “marker genes” that are exclusively expressed in one cell type. As a minimum, we accept that it is important to report the abundance of “marker genes” within liver and kidney, and we now do this (Fig. 6d; Supplementary Fig. 11f, h).

We have also explored an alternative approach, in which we avoid the potentially problematic concept of marker genes altogether. We analysed T>C conversion rates in those genes that were present in both liver and kidney RNAseq data. Using a differential expression analysis, we first defined genes as being liver-enriched or kidney-enriched. It would be unfair to compare crude T>C conversion rates within these gene sets within kidney tissue because unsurprisingly the kidney-enriched genes had, on average, higher expression within kidney tissue and we know that T>C conversion rate is expression-dependent, i.e. varies in proportion to the read count as cpm. Therefore, we then pulled genes at random from these three sets (liver-enriched and kidney-enriched) within defined cpm bins, so deriving two sets of genes that shared a similar abundance profile within kidney RNAseq data but differed in the extent to which they were expressed in liver tissue. After filtering out genes that exhibit strong labelling with 4sU, we compared T>C conversion rates between these gene sets within the kidney. We found that the liver-enriched genes exhibited higher rates of T>C conversion than did kidney-enriched genes in RNA-labelling groups (but not in the Cre-negative control group). We believe these data strengthen the fundamental claim of the paper.

We describe our approach in the supplementary methods (p27 – 28 of Supplement with tracked changes). We present these new analyses in Fig. 7d&e. We describe these results in the main text on p11 (of the tracked-changes version):

Within those genes that were present in both liver and kidney RNAseq data, we attempted to test whether T>C conversion rates were disproportionately higher in those kidney transcripts that were expressed at higher levels within liver. In a differential expression analysis, we first defined genes as being liver-enriched or kidney-enriched. It would be unfair to compare crude T>C conversion rates within these gene sets within kidney tissue because the kidney-enriched genes had higher average expression within kidney tissue and would therefore be expected to have higher rates of T>C conversion. Therefore, we pulled genes at random from these liver- and kidney-enriched sets within defined cpm bins, so deriving two sets of genes that shared a similar abundance profile within kidney RNAseq data but differed in the extent to which they were expressed in liver (Fig. 7d; Supplementary methods). Comparing T>C conversion rates, we found no consistent difference between rates of T>C conversion in liver-enriched and kidney-enriched genes within the kidney. However, when we repeated this analysis, first filtering out genes that were also labelled with 4sU (and therefore likely to be labelled by transferred nucleotides), we found that liver-enriched genes exhibited higher rates of T>C conversion than did kidney-enriched genes in RNA-labelling groups (Fig. 7e). Liver-enriched mRNAs were more strongly labelled than kidney-enriched mRNAs within the “Labelled in injury” (adjusted $p < 1^{-30}$) and “Labelled in health” (adjusted $p < 1^{-18}$) groups but not in the Cre -ve control group (adjusted $p > 0.05$). Taken together, these results are compatible with the transfer of both dissociated nucleotides and large mRNA fragments from liver to kidney.

Is there evidence for whole mRNA (fragments) in the kidney data?

One maybe more convincing analysis would be to look for reads mapping to 3'-ends of genes that are not expressed in the target cell type (kidney) at all but are abundant in the originating one (liver). To demonstrate, I conducted a quick-and-dirty search for genes that are highly expressed in hepatocyte datasets (median TPM > 5k) and lowly in kidney data (median TPM < 10) and plotted the data for these 28 genes (notably, only Mup3 is shared with the CellMarker list) in Review Attachment, Fig. 2. When looking at the number of T-to-C containing reads in the labelled kidney datasets, however, I found only marginal numbers

(max 3) converted reads in the labelled kidney datasets. Thus, I found no evidence that one of those transcripts was transported from liver (which of course does not exclude that others might have).

This is also an interesting idea and related to the point above. This approach is – by definition – biased against genes that might be transferred from liver to kidney (as it excludes genes that do not appear in kidney RNAseq datasets). Therefore, we prefer our approach above in which we use differential expression analysis to identify liver-enriched vs. kidney-enriched genes and then match for expression levels within the kidney.

I am also sceptical wrt. the discussion of the authors for distinguishing transferred transcripts (fragments) and dissociated nucleotides (p7ff.). The authors mention 478 mRNAs that were labelled by 4TU in the kidney under both basal and injury conditions, but not by 4sU (Table S11). When looking at their ARCHS4 expression, however, I found several that are very lowly expressed in hepatocyte/liver datasets as shown in Review Attachment, Fig. 3, which gives me low confidence that the respective reads stem from hepatocyte-expressed RNAs.

We acknowledge this is an important observation. We now present the abundance of these genes in liver and kidney tissue in our own RNAseq datasets, which are directly relevant, rather than in external datasets such as ARCHS4 (Fig. 7g).

The vast majority of these 478 mRNAs are expressed at reasonably high levels in liver tissue (median expression 33.2 cpm; 10th centile expression 3.3 cpm, meaning that 90% of genes are expressed at > 3.3 cpm in liver tissue and only 2 of the 478 genes were not present at all in the liver RNAseq data). It is not surprising that a small number of mRNAs were not present in abundance in liver tissue given that our approach is, as we fully acknowledge, unlikely to identify transferred genes with very high confidence.

Altogether, I don't see a robust way for distinguish between reads stemming from kidney and liver in this dataset besides studying the binomial probabilities for T-to-C conversions

as discussed above. This binomial probability should correspond to the effective 4sU concentration in the nucleus at transcription time and should arguably be much lower in the target (kidney) cells. So, if intact labelled RNAs are imported from liver into kidney, then they would have been transcribed with much higher 4sU concentrations in place and the 'density' of T-to-C conversions would be higher than for the respective RNAs transcribed in the kidney cells themselves. I doubt, however, that such an analysis (i.e., separating those two sets of RNAs or at least showing that there is more than one such set) is possible given the very low conversion rates in the kidney data and the complexity of such an analysis.

This is an interesting idea. We have tried to pursue it but came to the conclusion that this is not possible, for the reasons that you suggest.

To illustrate why this approach is unlikely to be successful, we examined the frequency of multiple T>C conversions within the same read. We analysed per-read T>C conversion data in our small RNA SLAMseq dataset. (We analysed the small RNA data rather than mRNA data because T>C conversion rates are computed across multiple reads mapping to a single 3'UTR in the SLAMDUNK analysis of mRNA data, and so per-read analysis of T>C conversion rates were not straightforward.)

Below, we plot the number of reads containing 0, 1, 2, 3... n T>C conversions, expressed as a proportion of all the reads that were derived by T>C conversions from a common 'parent sequence'. We plot these data for smallRNA in liver in the liver injury experiment:

We found that the vast majority of “labelled” reads comprise those with a single T>C conversion. Those containing two or more T>C conversions comprise under 1% of all reads derived from the common parent read and are an order of magnitude less common than those containing a single T>C conversion (note the \log_{10} scale on the y-axis). Therefore, it is highly likely that any reads containing two or more T>C conversions will appear so infrequently in the kidney that looking for reads with an unusually high density of T>C conversions will not provide a viable method for identifying transferred RNA.

Determining genes with significantly increased T-to-C conversions.

A central claim of the authors is that they detected relatively high percentages (5% under healthy conditions and 34% after acute hepatocellular injury) of, what they call, ‘labelled’

transcripts. To test for this, if I understood properly, the authors applied a two-group FDR-controlled beta-binomial test to the conversion rate output from SLAMDUNK (and from their own smallRNA pipeline discussed below) to identify genes that are “labelled” in their terminology. I have several questions in this regard:

- As far as I know, the beta-binomial test was proposed for comparing count data, so the authors should explain why this test is appropriate for their purposes and extend the respective description of how the test was conducted in the methods section. Also, the respective data tables including the output from the countdata package (p-values, and adjusted p-values as well as normalized and fold-change values) as well as the abundance (in TPM) per gene should be provided (in the current version, I just found lists of genes that passed the test).

The beta-binomial test has been used by others for exactly these purposes, and we replicated this published approach (e.g. see Matsushima et al. (2018), Development, <https://pubmed.ncbi.nlm.nih.gov/29945865/> or Matsushima et al., (2019), Nature Protocols, <https://pubmed.ncbi.nlm.nih.gov/31243395/>). The justification for using this statistical test is that T>C conversion is a binomial process and therefore the probability of any given T being converted to a C can be fitted with a beta-binomial distribution. We provide the full output from the beta-binomial tests as suggested (Supplementary Data Files 5 & 9). We have also reported the outcomes of our beta-binomial tests more comprehensively as graphical data (in Supplementary Figs. 11 – 14) which include plots of gene abundance in different experimental groups. (The main conclusions are not altered.)

- Additionally, the authors should confirm that this method does not report false positives when applying to sets of genes that are unlabelled in liver data (see marker gene discussion above) or by using in-silico simulation data with given (fixed, low) conversion rates/sequencing error. Did the authors apply this approach to subsets of Cre-neg replicates or compare Cre-neg and 4TU-neg data?

We found that this method did not report any positives in the kidney smallRNA SLAMseq data, which is one line of evidence against it reporting false positives in unlabelled datasets.

In response to this comment, we conducted some *in silico* simulations as suggested. We first generated a dataset with a distribution of gene abundance (cpm) and number of Ts sampled per gene (“coverage-on-Ts”) that broadly recapitulated our real experimental data. To do this, we computed the mean cpm and coverage-on-Ts for each gene in our experimental RNAseq data from kidney tissue in the Cre -ve group in the liver injury experiment.

From this dataset, we made repeated samples, simulating gene-wise T>C conversions with a defined “true” overall T>C conversion rate. For each repeat, we sampled 1000 genes at random; we set the number of Ts sampled (“coverage on Ts”) per gene at random within a window centred around the mean value for that gene within all Cre -ve libraries and extending +/- 20%; we computed the number of T>C conversions per gene, setting a conversion rate at random within a window extending +/- 30% from our defined population “true conversion rate”.

We used this sampling procedure to generate two groups of simulated libraries, with n libraries within each group. We set different “true conversion rates” for each group.

We then performed the beta-binomial test on these simulated data and computed the proportion of genes that were detected as being “labelled” in simulated group 2 (c.f group 1). We performed this whole process 8 times, to generate confidence intervals around these proportions.

We used this *in silico* simulation to explore two scenarios. First, we set a low rate of T>C conversions that was identical in both groups (5×10^{-4}). In this scenario, we detected zero “false positives” over a range of group sizes ($n = 2 - 10$). That result provided reassurance that this method does not report false positives.

Second, we explored how group size affected the performance of the beta-binomial test, to answer the comment immediately below. (See below for these results.)

- Also, in the text it is mentioned that (in the initial experiment) 0.6% of all analysed genes in the (labelled) kidney QuantSeq dataset were considered as labelled. What was the abundance of those genes? Did the probability to be labelled correlate with a gene's abundance? What fraction of reads were labelled and how does this compare to the liver dataset? How did the dataset with the high conversion rate (cf. Fig2B) influence this number?

There were very few replicates ($n = 3$ per group) in that initial experiment, making this sort of interpretation challenging. Hence why we proceeded to a more definitive experiment with larger group sizes ($n = 6$ per group).

We reasoned that when there are very small differences in T>C conversion rate between groups, the beta-binomial test is likely to be sensitive to group size. We have now tested this assumption, using *in silico* simulation data as described immediately above. We defined conversion rates that were very low, but that differed between two groups (5×10^{-4} and 8×10^{-4} respectively – very approximately what we observed in kidney in our pilot experiment). We found that as group size increased, so did the proportion of genes that were identified as being labelled by beta-binomial testing.

For this reason, we performed our more in-depth characterisation of labelled genes in the liver injury experiment ($n = 6$ per group).

Technical description of the Quantseq analysis.

The description of the Quantseq data analysis is insufficient in my opinion and needs to be extended:

Thank for you for requesting these additional details, which we include in the revised methods sections (see p21-22 in the main document and p24-25 in the Supplement; tracked-changes versions).

- What 3'-end annotations were used and how were they calculated?

The 3' end annotations used were as per the original publication describing SLAMSeq technology (<https://www.nature.com/articles/nmeth.4435>). The Bed files are at: <https://www.ncbi.nlm.nih.gov/geo/query/acc.cgi?acc=GSE99970>.

- What version of SLAMDUNK was used? What commandline was used?

Version 0.4.3.

```
slamdunk map -t 18 -r  
mmu_GRCm38.92_amerseq/annotation_organism_ercc_sirv.fa -5 12 -a 4 -n 100 -ss  
-o 2935663/tmp/ sample_sheet.tsv
```

```
samtools view -H 2935663/tmp/R1.fastq_slamdunk_mapped.bam | sed 's/ID:[0-  
9]/ID:24/' | samtools reheader - 2935663/tmp/R1.fastq_slamdunk_mapped.bam >  
2935663/output/R1.fastq_slamdunk_mapped.bam
```

```
slamdunk filter -t 4 -b  
mmu_GRCm38.92_amerseq/annotation_organism_UTR.bed -o 2935687/output/  
2935663/output/R1.fastq_slamdunk_mapped.bam
```

```
slamdunk snp -t 6 --var-fraction 0.2 -r  
mmu_GRCm38.92_amerseq/annotation_organism_ercc_sirv.fa -o 2935711/output/  
2935687/output/R1.fastq_slamdunk_mapped_filtered.bam
```

```

slamdunk count -t 4 -b
mmu_GRCm38.92_ameses_slamseq/annotation_organism_UTR.bed -r
mmu_GRCm38.92_ameses_slamseq/annotation_organism_ercc_sirv.fa -o 2935735/output/
-s 2935711/output --max-read-length 101
2935687/output/R1.fastq_slamdunk_mapped_filtered.bam

alleyoop collapse -t 4 -o 2935735/output/
2935735/output/R1.fastq_slamdunk_mapped_filtered_tcount.tsv

alleyoop rates -t 4 -o 2935735/output/ -r
mmu_GRCm38.92_ameses_slamseq/annotation_organism_ercc_sirv.fa
2935687/output/R1.fastq_slamdunk_mapped_filtered.bam

alleyoop utrrates -t 4 -o 2935735/output/ -r
mmu_GRCm38.92_ameses_slamseq/annotation_organism_ercc_sirv.fa -b
mmu_GRCm38.92_ameses_slamseq/annotation_organism_UTR.bed --max-read-length
101 2935687/output/R1.fastq_slamdunk_mapped_filtered.bam

alleyoop tcperreadpos -t 4 -o 2935735/output/ -r
mmu_GRCm38.92_ameses_slamseq/annotation_organism_ercc_sirv.fa -s 2935711/output --
max-read-length 101 2935687/output/R1.fastq_slamdunk_mapped_filtered.bam

alleyoop tcperutrpos -t 4 -o 2935735/output/ -r
mmu_GRCm38.92_ameses_slamseq/annotation_organism_ercc_sirv.fa -s 2935711/output -
b mmu_GRCm38.92_ameses_slamseq/annotation_organism_UTR.bed --max-read-length
101 2935687/output/R1.fastq_slamdunk_mapped_filtered.bam

```

- The read mismatch rates per read position should be plotted (e.g., SLAMdunk output) and provided for all datasets.

We have now done this (as per comments above re: MultQC data in supplement).

- Did the authors observe increased mismatch rates at the read ends and consider clipping the read 5'-ends? If yes, how many bases were clipped?

The first 12nt are automatically removed at the 5' end of the reads. This is described in the command line 1.

- What var_fraction (given the low overall conversion probabilities) was chosen to exclude SNPs?

The var_fraction chosen to exclude SNPs was set to 0.2 (see command line 4).

- SLAMDUNK provides many useful QC statistics (e.g., conversion rate per read or T>C conversions over UTR positions) that can be used to determine biases in the data. Those should be added to the supplement.

We have now done this (as per comments above re: MultQC data in supplement).

- Did the authors consider controlling duplication rates with UMIs?

We did consider this but in the end elected not to use UMIs. For SLAMseq, the benefit of UMIs is limited since SLAMDUNK only analyses the ratio of T>C conversion and this is on average not affected by PCR duplicates. UMIs are in general more relevant if the read depth is high compared to the input amount and/ or the input amount used to generate the libraries is very low; neither of those factors were at play in our study.

Technical description of the smallRNA analysis.

The smallRNA results of this study are also of concern. The high abundance of MicroRNAs in exRNA (as reported by many studies), for example, and their increased stability when compared to mRNAs should in principle make it easier to detect them if they are indeed transported from liver to kidney cells. The authors explain not finding labelled smallRNAs in kidney data by "insufficient sensitivity" (not sure what this means) of their approach, but I don't think they have presented enough evidence to exclude technical reasons related to their analysis pipeline. I have several questions/comments in this regard:

- How was the proposed pipeline evaluated? Was synthetic (simulated) data used to validate its proper function?

We know that robust RNA labelling was achieved in liver tissue, and this provides a nice positive control with which to confirm that our pipeline was working. We did see, as expected, increased labelling in the liver sRNA dataset.

We deliberately designed a pipeline in which T>C conversions were quantified first in a mapping-free approach. Mapping small RNAs can be problematic, but this approach meant that we were able to confidently quantify T>C conversion in the whole sRNA dataset without concerns about accurate mapping.

However, in response to your comment, we re-evaluated our approach to ensure that our original mapping step did not impair the sensitivity of detecting T>C conversions in reads mapping to miRNA (or indeed any other given class of small RNAs). In our original analysis, reads were mapped sequentially to different small RNA genomes (rRNA then tRNA then miRNA and so on, with the final step being mapping of any reads so far unmapped to the whole genome).

As an alternative, we re-analysed our sRNA data, mapping once to the whole genome in a single mapping step. This might increase the sensitivity for detecting T>C conversions, if reads that map to a labelled small RNA (e.g. miR-122 that we think almost certainly is being transferred from liver to kidney) were being mapped to alternative genomic loci in rRNA or tRNA. We present this in Supplementary Fig 15 (and provide the methods in the Supplement, p26 in the version with tracked changes). We again did not detect labelled small RNA within the kidney.

We have not used simulated *in silico* data to test our pipeline because we think the *in vivo* positive control data (i.e. liver RNAseq data) is a more convincing test.

- Does the pipeline consider 3'/5'-isoforms and untemplated additions (tailing) of small RNAs?

No, this is a limitation and we acknowledge that in our revised discussion (new Limitations section, bottom of p16 of the tracked-changes version):

Our small RNA pipeline did not account for some features that may be important when quantifying and mapping T>C conversions. For example, it did not consider RNA isoforms and untemplated additions. Future work attempting to use SLAMseq to track mobile small RNAs may need to account for these phenomena, as well as addressing the issue of poor sensitivity that we encountered.

- What duplication rates were observed by the authors? Why were no UMIs used for deduplication?

Duplication rates were ~90% (see new MultiQC files in the data supplement). Our reason for not using UMIs is given in response to a comment above. We do not think our approach is likely to introduce a bias because there is no *a priori* reason why a read containing T>C conversions should be amplified more (or less) than any other read.

- How were pre-processed reads mapped to the respective reference sequences? (bowtie2 command line)

We have added the command line to our methods (for both the main sRNA pipeline and our sensitivity analysis) – see p22 of main document and p26 of the Supplement (tracked-changes versions).

Minor comments

- Fig 1B: add box and jitter-plots (to avoid overplotting) and test for significance.

We have now done this.

- Page 5: reference “supplemental Tables S4 & S5” should be “supplemental Tables S5 & S6”? All table/figure references should be checked.

Thank you for spotting; all table and figure references have been updated and double-checked in the revised manuscript.

- Fig 2G: what are 'unmapped' small RNAs? How was the T-to-C conversion rate of those estimated?

Unmapped RNAs were reads that were not mapped to any of the small RNA genomes (rRNA, tRNA, miRNA, pRNA etc.) or to the whole genome. Because our pipeline quantifies T>C conversion first and then maps to the genome, we can report T>C conversion rate for unmapped reads.

- Fig 1: subfigure labels are wrong in the caption.

We cannot find this error in the Fig 1 legend but we did double-check and revise all of our figure legends.

- Fig 4E: Besides clustering of the "Labelled in injury" data, the PCA also shows clear clustering of the control datasets along PC2. The authors should discuss and investigate (can these differences partly be explained due to transcriptional changes induced by the AAV8-TBG vector?).

This is a very good point. We have explored this with some further group-wise differential expression analysis (Supplementary Fig. 8). In liver, we found that the combination of 4TU and Cre exposure resulted in activation of injury-response pathways, including the known target of 4sU exposure, p53. These findings provide important context for our results overall, meaning that even in the "RNA labelling in health" group, there was a low level of hepatocyte injury. In kidney, we found that 4TU and Cre exposure did not induce any significant differential gene expression, providing further evidence that RNA labelling occurred within hepatocytes and not within kidney cells.

We present these results in the main text on p7 of tracked-changes version, paragraph 3:

The combination of AAV8-TBG-Cre and 4TU treatment induced global transcriptional changes in the liver, with upregulation of injury response pathways including p53 signalling: a known consequence of 4TU-induced cellular toxicity (Supplementary Fig. 8a-f).²⁴ Therefore, the “Labelled in health” group, whilst relatively healthy compared to the “Labelled in injury group”, exhibited low levels of liver injury induced by 4TU exposure. In contrast, the labelling protocol did not induce any significant global transcriptional changes within the kidney, providing further evidence against any off-target labelling of kidney RNA with our protocol (Supplementary Fig. 8g-h).

- I am missing detailed mapping statistics for the smallRNA datasets of the main experiment (cf. Fig 2I for the initial experiment)

These are now included (Supplementary Fig. 12c,e).

- Software, database and annotation versions & commandlines should be added to the Supplement.

Thank you, we now include these.

- Table S3: how were conversion rates calculated?

The conversion rates were simply summarised over the entire dataset for that experimental group.

- Table S12: Versions of the annotation datasets are missing.

Thank you, we now include these (Supplementary Table 6).

- The sample sheet I was given during the review process should be published alongside the study data.

Thank you, we now include this (as Supplementary Data File 2).

- Where can I see T-to-C conversion rates per dataset for the main experiment?

As mentioned above, these are now presented in Supplementary Figs. 11-14.

Reviewer #3 (Appendix):

Report on my small re-analysis of parts of the Quantseq data.

I have downloaded the datasets from ENA and, thanks to the sample sheet provided by the authors, did a quick and dirty analysis of all Quantseq/labelled and unlabelled datasets:

- I used the SLAMDUNK nextflow pipeline (<https://nf-co.re/slamseq/1.0.0/>)

- I set the following parameters:

```
nextflow run nf-core/slamseq --fasta ${REF} --bed ${BED} -- trim5 5 --var_fraction 0.1 --
read_length 100 --skip_deseq2 --input slamdunk_sample_sheet.tsv -profile singularity
```

- My reference sequence was GRCm38 (mm10)

- My BED file with 3'-end annotations (SLAMDUNK counting windows) was created from gencode.vM21 annotations. I calculated the (spliced) 3'-end intervals of length 1000bp for each annotated transcript, grouped by gene and flattened the intervals (i.e., merging overlapping ones).

- I trimmed 5 bases from the 5'-end as I observed reduced sequencing qualities in some datasets in a pre-experiment.

- I set the var_fraction parameter to 0.1 due to the low conversion rates in this dataset.

Looking at the multiqc summary, SLAMDUNK reported no converted UTRs for the kidney samples and very low conversion rates per read, well within the range of sequencing error (compare, e.g., T>C with C>T and A>G with G>A in for the pink/red samples), Fig 4.

Fig 4: SLAMDUNK QC plots. Top row: Conversion rates per UTR and T>C conversions over read positions. Bottom row: Conversion rates per read for plus and minus strand reads.

Colors: dark-blue: liver/labelled, cyan: liver/labelled_APAP, red: kidney/labelled, pink: kidney/labelled_APAP)

When I added all datasets from the 'unlabelled' category, I observed high mismatch rates in the unlabelled/liver but not the unlabelled/kidney data, Fig 5.

We provide our own data in similar format (i.e. multiQC output from SLAMseq analysis) below. We agree that T>C conversion rates in kidney samples are all very low, but there was a statistically significant difference between RNA labelling and control negative groups. We also agree that overall nucleotide conversion rates in negative control groups were higher in liver samples than in kidney samples. This is not particularly surprising, given that the set of genes expressed in those tissues is likely to differ substantially between liver and kidney. We do not think this invalidates any of our analyses; we never make direct comparisons between T>C conversion rates in different tissues.

Reviewer #3 (Remarks on code availability):

I tried to use the small RNA pipeline myself, but it does not run out of the box as it contains many local path and infrastructure (e.g., HPC configuration) references.

After fixing those paths, I manually ran "0b_pull_smallRNA_fasta.R" to download the reference sequences, however, the script failed some datasets could not be downloaded

completely. For example, Rscript ../scripts/0b_pull_smallRNA_fasta.R RNA_biotypes_source_split_1.csv resulted in the following error: 'Error: lexical error: invalid char in json text [...]', and the created miRNA.fa file contained only 97 sequences. After rerunning the command, the same error occurred but the file now contained 697 entries (that were appended to the already existing file!). After deleting the file and running the script a 3rd time, it finished w/o error and contained 1860 sequences. Some other datasets also failed with the error above or with: 'Error: parse error: premature EOF'.

At this point I stopped testing the smallRNA pipeline for now.

I strongly recommend fixing those issues and testing the pipeline thoroughly with simulated data. To increase pipeline reusability, I suggest publishing a respective Docker/apptainer image and to upload the RNA reference sequences as used in this study (except for the mouse genome sequence) to a public repository (Zenodo) to enable reproducibility of the analysis.

Thanks for pointing out all these errors and apologies this proved so troublesome. Our intention was to be as transparent as possible about our methods and to make our code publicly accessible as part of this. It is not our intention to publish a refined pipeline; we lack the infrastructure and resource to maintain such a pipeline. We weren't sufficiently clear about that in our original manuscript and have revised accordingly. Our revised Methods section now states (p22 of the tracked-changes manuscript):

*Our code is publicly available
(<https://github.com/robertwhunter/smallSLAM>;
<https://zenodo.org/doi/10.5281/zenodo.11083131>), albeit not in the form of
a robust and readily reproducible pipeline.*

As a side note, I wanted to let the authors know that in a pre-experiment I did, I noticed that their smallRNA dataset ERR13245034 looks very different from all other smallRNA datasets I looked at in the QC plots. It showed elevated T/C conversion rates when compared to the other liver datasets and uniform distribution of non T>C conversions over read and the UTR positions. Can the authors explain why?

These data were from sample “L2”: one mouse in the group of three that received the RNA labelling protocol (i.e. AAV8-TBG-Cre then 4TU). The RNA labelling (i.e. T>C conversion) was stronger in this mouse than in the other two mice in that group. This probably just reflects the biological variation of our RNA labelling protocol; there were no obvious reasons to explain why labelling was stronger in this particular mouse. As detailed in response to reviewer 2, we now show all data for individual replicates to make the extent of between-replicate variation transparent.

REVIEWER COMMENTS

Reviewer #1 (Remarks to the Author):

The authors provided a reasonable response to my comments, however, some suggested experiments that should have raised interest for the paper and better detailed the destination of the identified liver-derived RNA (RNA detection within urine and within isolated tubular cells) were not performed. This was justified by the complexity of the experiments, low sensibility and time. The message remains interesting, but its physiological relevance is less clear.

Thank you very much for taking the time to review our response. We completely agree that the physiological relevance of liver-to-kidney RNA transfer is still to be established. We have amended our discussion to make this limitation transparent and explicit.

On p16, we have added the following text to the end of our limitations section:

Finally, our work does not address the physiological consequences of any RNA transfer. The extent to which this phenomenon impacts on kidney physiology remains an important, unanswered question.

We have also amended our abstract, adding the following concluding sentence:

There were important limitations: we could not confidently identify transferred RNA transcripts at the single-gene level and we did not assess the physiological consequences of any RNA transfer.

Reviewer #1 (Remarks on code availability):

The data within code can be used by the community

Reviewer #2 (Remarks to the Author):

The authors have made the suggested changes and addressed all the comments.

Thank you.

Reviewer #3 (Remarks to the Author):

I thank the authors for addressing all my comments, for providing additional analyses and QC data and particularly for rephrasing the limitations sections. I agree that the made changes strengthen the paper.

Analyzing this dataset seems challenging and overall, my confidence in the presented data remains rather low. The revised limitations section and various adaptations in the manuscript, however, now reflect the remaining doubts about the found evidence for the passage of intact/functional mRNA molecules from liver to kidney cells in mouse. As the authors address a relevant research question, I thus believe that the created datasets, the presented results and the developed methodology will be of interest to the community.

Major comments

A fundamental question of my previous review was whether the observed T-to-C mismatches in the kidney data originate from 4sU incorporation in liver cells or whether they are simply sequencing errors or have other causes. The now provided SLAMDUNK multiQC reports are important for judging on the overall data quality of the Quantseq datasets. The `injury_expt_liver_mRNA_multiqc_report` shows expected T-to-C conversion rates over the 3' ends for the labeled samples, although there seems to be an artefact at the read ends (peak in "T>C conversions over reads" plot at pos 87-88). I observed the same artefact in my small re-analysis of the data in the previous review round.

The `injury_expt_kidney_mRNA_multiqc_report` show the same artefact but much stronger pronounced as overall "conversion" rates are much lower here. As this peak is observed in all samples (but not in the non T-to-C conversion plots) it is likely a direct consequence of the applied sequencing/bioinformatics protocol. As it cannot be excluded that this influences overall results in the kidney data (given the low observed conversion rates), the cause of this artefact should be investigated (possibly read ends need to be

trimmed before running SLAMDUNK or GRANDSLAM? The GRANDSLAM mismatch plots may also be helpful to investigate this).

Thank you, once again, for such a thorough critique of our work. It has been so helpful.

This artefact is rather odd. We have performed some additional analyses to characterise it and to establish the extent to which it might impact on our conclusions.

It is clearly an analysis artefact. When the reads are trimmed and re-analysed on the SLAM-DUNK pipeline, the artefact persists but shifts to what are now the last two read positions (see Reviewer Figs R1A & R1B in the appendix below). Interestingly, the same artefact is evident in the GRANDSLAM mismatch plots (Reviewer Figs R2A & R2B).

Lexogen process a lot of SLAMseq data. When we spoke to them about this, they comment that they have observed a similar artefact on other experiments (i.e. from other investigators). They believe it to be caused by the alignment method.

The artefactual excess of T>C conversions will be carried through into the rest of the SLAMDUNK (or GRANDSLAM) analysis, which will have a small effect on the final results. However, we think this effect is likely to be negligible and will not alter our main conclusions for two reasons. First: this artefact is present in all samples – i.e. affects all experimental groups equally. The between-group differences are evident and consistent at the lower read positions (Reviewer FigR3). Second, it contributes to a tiny proportion of all T>C conversions, even in the kidney data (Reviewer Figs R4A & R4B).

The injury_expt_kidney_mRNA_multiqc_report report shows low but possibly increased (as compared to the controls) T-to-C conversion rates over the 3' ends for the labeled samples. The “conversion rates per read” plot, however, also shows, e.g., C>T conversion rates on the plus strand (likely sequencing or RT errors) that are even higher than the observed T>C conversion rates for some labelled datasets and it seems as if overall mismatch percentage in labelled samples is slightly higher than in unlabeled ones. This is relevant as all downstream analyses take only T-to-C conversions into account and don't correct for sequencing errors as the authors confirmed (except for simple per-base quality filtering in

SLAMDUNK). In this regard, it was very helpful that the authors repeated their analysis with GRANDSLAM which takes T-to-C sequencing errors into account to some extent (cf. <https://github.com/erhard-lab/gedi/wiki/GRAND-SLAM#more-explanations-on-grand-slam> for a discussion). To better understand the results of this analysis, it would be helpful if the authors could provide the binomial estimates from those experiments (i.e., all *binomEstimated.tsv files possibly combined in one table) and add the overall ntr plots (*ntr.png) to the Supplement as I assume that Sup. Fig 6 was created from the ntr values per gene/utr?

Could GRANDSLAM reliably estimate the conversion rates (reported in the Status column)? How do the estimated conv_new measures compare between liver and kidney cells?

We provide the additional GRANDSLAM outputs requested in the data supplement.

We have now included the binomEstimated.tsv files, combined into a single supplementary table (new Supplementary Data File 13). We also include the ntr.png plots (Supplementary Fig 7 revised to include new panels g & h). For ease of reference, we also replicate these in the appendix below as Reviewer Figures R5A & R5B.

The conv_new measures were broadly similar in liver and kidney. The Status reported in the *binomEstimated.tsv output shows that GRANDSLAM was not reliably able to estimate conversion rates in kidney tissue. We don't think this is necessarily surprising. There are clear between-group differences in T>C conversion rates in kidney tissue, as detected by both the SLAM-DUNK and GRANDSLAM pipelines.

Minor comments

- table "509530_1_data_set_10016079_sn2znx" (containing study accessions and other metadata) does not contain data for s10 and s34?

These libraries were never actually included in our analysis for this paper because they were from a mouse that was not included in the final study. (We excluded the mouse based on suboptimal tissue perfusion at the time of cull. We inadvertently included the tissues in the batch that were used for RNA extraction, library preparation

and sequencing – hence why the sequencing files were erroneously uploaded to ENA.) Thanks for spotting the error. They should not have been included in that ENA submission. We have contacted ENA to request that these libraries be removed. If that is not possible for any reason, we will amend the metadata to make it clear that these libraries are not part of this study.

- In the discussion, the authors write “Our main finding is that RNA is transferred en masse from the liver to the kidneys in mice”. In light the discussion above and the acknowledged limitations of this study I encourage the authors to rephrase this sentence accordingly.

Our revised sentence reads:

Our main finding is that liver-derived RNA is apparently detectable within the kidneys.

Response appendix: figures for reviewer 3

Reviewer Fig R1A. Excerpt from original multiQC analysis of all kidney libraries from the QuantSeq SLAMDUNK liver injury experiment. An analysis artefact is observed, showing inflated T>C conversion rates at read positions 87-88.

Reviewer Fig R1B. Excerpt from repeat multiQC analysis of all kidney libraries from the QuantSeq SLAMDUNK liver injury experiment, having first trimmed 6 bases from the 3' end of all reads. The analysis artefact persists, but has shifted to read positions 81-82.

*Reviewer Fig R2A. T>C conversions per read position, extracted from GRANDSLAM analysis of liver libraries. Data were taken from *mismatchdetails.tsv files, and report averages per experimental group.*

Reviewer Fig R2B. As for fig R2A, but for kidney libraries.

*Reviewer Fig R3. T>C conversion rates in kidney libraries from the liver injury SLAMseq experiment. Data extracted from *.tperreadpos.csv files to replicate the plot in multQC analysis, but re-plotted so that libraries are colour-coded according to their experimental group. This makes it clear that there are consistent between-group differences in T>C conversion rates at lower read positions (i.e. excluding the artefact at positions 87-88).*

Reviewer Fig R4A. T>C conversion rates in liver libraries from the liver injury SLAMseq experiment. Data extracted from *_tcrperreadpos and then used to compute overall conversion rates from all read positions or having first excluded read positions 87 and above.

Reviewer Fig R4B. As for Fig R4A, but for kidney libraries.

*Reviewer Fig R5A. NTR (new-to-total RNA ratio) estimates from GRANDSLAM analysis of liver libraries from the liver injury SLAMseq experiment. Data extracted from *ntrstat.tsv files in order to replicate .ntr.png plots. The error bars show upper and lower bounds of the confidence interval. Each experimental group was compared to the same three 4sU-negative control libraries (represented for context in each of the four panels).*

Reviewer Fig R5B. As for Fig R5A, but for kidney libraries.

REVIEWER COMMENTS

Reviewer #3 (Remarks to the Author):

I thank the authors again for addressing all my comments, for providing additional analyses and QC data and for revising the main manuscript.

- Re. the observed T>C artefact: Are you sure you trimmed the right end of the reads? Anyways, I assume that you applied GRANDSLAM to the BAMs created by SLAMDUNK and would therefore of course observe the same artefact in those QC plots (Fig R2A+B) which are, however, cleaner than the slamdunk/multiqc plots in my opinion. I agree that this could possibly be a mapping artefact from NextGenMap and the authors have nicely shown that overall impact on the analysis is negligible.

- Re. GRAND-SLAM data: The binomial estimates indeed confirm a slightly higher p_{new} for labeled kidney data (see attached plot) and adding this table to the supplement was useful in my opinion. I agree that the fact that GRANDSLAM could not reliably estimate conversion rates in kidney data is explained by the very small number of actually detected T-to-C conversions.

- Minor comment re. suppl fig 7: There is a missing full-stop and an extra one after "g,h" in the revised figure caption.

I have no further comments or suggestions.

Thank you – once again – for subjecting our work to such careful, rigorous review. The final paper is stronger because of your input; thank you. We are pleased that we were able to satisfy your residual concerns. Thank you for spotting that typographical error in supplemental FigS7.

In the manuscript entitled "**SLAMseq reveals transfer of RNA from liver to kidney in the mouse,**" the authors have evaluated the transfer of RNA from the liver to the kidney in health and after acute hepatocellular injury. The authors have labelled the hepatocyte RNA and observed increased mRNA labelling in kidney transcripts after hepatocellular injury. The work is found original. However, the following concerns are there:

- To confirm the development of paracetamol-induced acute liver injury, only the plasma alanine transaminase (ALT) levels have been checked in the present study. Authors should also evaluate the changes in liver function markers, including aspartate aminotransferase (AST), alkaline phosphatase (ALP), and total bilirubin.
- The authors have mentioned that the mRNAs were enriched in mitophagy, autophagy or ferroptosis. A table can be provided to classify the mRNAs explicitly enriched in each mechanism.
- The authors have performed the GO analysis. However, the enriched biological process (BP), cellular components (CC), and molecular functions (MF) have not been added anywhere in the manuscript, which are the critical parameters of GO analysis. The authors should incorporate the same in the manuscript.
- The title indicates the transfer of RNA from the liver to the kidney. It can be modified for clarity by incorporating the condition where this transfer occurs, either healthy or liver injury. Similarly, the introduction can be improved for clarity and better understanding.
- The authors should provide better representative images for H&E staining.
- For Figure 1D, triplicate blots should be provided in the supplementary files.
- The authors should also elaborate on the potential implications of these findings in the discussion. Comparisons to existing studies are made, but these could be expanded to more critically evaluate how the current findings align or contrast with previous work. Discuss any discrepancies or confirmations between your findings and those previously published.
- The authors observed a small to no increase in labelled small RNA in acute hepatocellular injury compared to normal. What could be the potential reason that only the transfer of labelled mRNA from the liver to the kidney has been increased and not of small RNA? Please discuss.
- For better understanding, please define the abbreviations at their first appearance in the manuscript.
- There are a lot of grammar and sentence construction mistakes. The manuscript would benefit from proofreading to correct occasional grammatical errors and improve sentence structure for better readability.

Review for "SLAMseq reveals transfer of RNA from liver to kidney in the mouse" by Hunter et al., 2024

In the manuscript "SLAMseq reveals transfer of RNA from liver to kidney in the mouse", Hunter et al. investigate the transfer of RNA between liver and kidney cells in mice by metabolic RNA labelling. Briefly, mice (healthy or after induced acute hepatocellular injury) are exposed to 4-TU which is converted to 4sU in hepatocytes by specific expression of HA-UPRT (cf. SLAM-ITseq, Matsushima et al., 2019).

Active 4sU is incorporated randomly into nascent hepatocyte transcripts. RNA from liver and kidney is extracted and subjected to the SLAM-seq protocol. Finally, RNAs are sequenced by two protocols: 3'-end sequencing (Quantseq, targeting polyadenylated RNAs) and smallRNA sequencing. The resulting datasets are then analysed for evidence of liver-originating RNA in kidney data that could have been transferred as exRNA between tissues.

The study approach appears reasonable as (i) tracking exRNA with SLAM-seq was successfully done before (cf. e.g., Sharma et al., 2018), (ii) it was reported before that exRNA vesicles may contain mRNAs (or fragments thereof) and (iii) it was reported that, e.g., miR-122 that was secreted from (injured) liver cells entered kidney cells and reduced expression of Epo therein (cf. Rivkin et al., 2016). While the authors are targeting a relevant and significant research question, I have considerable doubts that the presented data supports the claims made in the paper.

Major comments

The main claims in this paper are based on the bioinformatics analysis of the SLAM-seq Quantseq data (I will discuss the smallRNA data below), which relies on strict hepatocyte-specific expression of HA-UPRT (as the authors acknowledge and confirm in Figure 1), but also on the accurate interpretation of T-to-C conversion data and filtering of alternative sources for such observed mismatches (background conversion rates, sequencing error rates, SNPs, misaligned reads, duplicated reads, etc.). This is particularly important in this study as the reported T-to-C conversion rates in the presumed target tissue (kidney) are very low and close to reported sequencing error rates. In the current version of the paper I am, however, missing a stringent analysis in this regard.

- The current set of Figures makes it hard to judge on the data quality as key statistics such as the mismatch-rate per read position are missing. In Figure 2B, for example, I am missing more detailed mismatch statistics per dataset and read positions (as read ends are often enriched in unspecific mismatch rates, cf. Quantseq comments below).

- Also, the shown T-to-C box shows a low median and a large IQR. In my re-analysis of (parts of) the dataset (see below), I found out that one of the replicates has a considerably higher conversion rate than all others (see below).
- In Figure 3A, the mismatch profiles show very low T-to-C conversion rates that are similar to unspecific (sequencing error) mismatch rates (e.g., C-to-T).
- Along these lines, testing for significantly higher T-to-C conversion rates needs to be stratified/normalised by the dataset specific sequencing error rate (e.g, in Table1). My small re-analysis of parts of this dataset showed varying levels of sequencing errors (e.g., in the unlabelled datasets, see below) but I don't see where the authors correct for those.
- I am also missing an analysis of the biological replicates, e.g., how reproducible were T-to-C conversion rates between replicates?
- Also, in Fig 5+6: the observed increase in T-to-C conversion rate in liver in the paracetamol experiment could be due to a higher 4TU concentration in the respective cells. The fact that a similar increase is observed in the kidney could indeed be explained by increased liver>kidney transfer of respective RNAs but also simply due to unspecific conversions in the kidney resulting from higher 4TU levels. Indeed, Cre -ve shows higher conversion rates than 4TU -ve. How does this fit into the author's model?
- The authors report a positive correlation between T-to-C conversion rates and gene abundance and explain this (e.g.in the main text describing Figure 2C) by: "The T>C conversion rate was higher in more abundant transcripts, consistent with more abundant RNA species being transcribed at a higher rate". Transcript abundance, however, is determined not only by transcription rate but also decay rate and other factors. Genes with low transcription rates but high stability are also highly abundant. The same inaccuracy found on page 7, line 176. In the present setting, I assume that the detection of (intact) liver RNAs in the kidney data would particularly depend on the stability of those transcripts in kidney cells, it would thus be more intuitive to instead correlate conversion rates with half-lives (instead of abundances) from published mouse datasets or meta-analyses (e.g., Agarwal&Kelley, 2022).
- One recommended quality control step would be to calculate the binomial probability for observing T-to-C mismatches in old (mainly due to sequencing/mapping errors, unfiltered SNPs, etc.) and new RNA (due to the SLAM-seq conversion) for each sample as done, for example, by GRAND-SLAM (<https://doi.org/10.1093/bioinformatics/bty256>). The authors could compare these rates between replicates and sample groups. The binomial probability (if it can be fitted for these low conversion rates) would serve as a proxy for the effective 4sU concentration in the respective cells be considerably higher in liver cells.

Is there some ‘liver’ signature in the data?

The authors also try to find some kind of “liver” signature in the data to strengthen the claim that the observed labelled (if indeed) reads originate from liver. They claim that “Labelling was evident in kidney transcripts mapping to known hepatocyte marker genes, to a greater extent than those mapping to markers of other cell types.” I am, however, sceptical whether selecting hepatocyte marker genes from CellMarker 2.0 to show that hepatocyte-originating RNA (fragments) are imported into kidney cells is the right approach. Marker genes are typically identified by differential gene expression (DGE) which is based on a statistically significant difference between, in this case, hepatocyte data and other tissues but does not imply that there is no kidney expression at all. Thus, if the probability to be identified as ‘labelled’ by the authors approach depends on a transcript’s abundance then this analysis would not be insightful as it would depend on the abundance of the respective cell marker genes in the target (kidney) cell type which the authors did not report on. As a quick check, I queried the ARCHS4 (<https://maayanlab.cloud/archs4/>) mouse dataset and plotted the expression in bulk-RNAseq experiments for marker genes listed in Table S11. I included only datasets that were sequenced on a HiSeq sequencer with more than 5Mio aligned reads and excluded the ones with a ‘singlecellprobability’ ≥ 0.5 . I queried the "series_id", "characteristics_ch1", "extract_protocol_ch1", "source_name_ch1", "title" metadata fields for the search terms “Hepatocyte” and “Kidney” and assigned the datasets to respective sets resulting in 4131 kidney and 1086 hepatocyte datasets. Repeating this analysis with using “Liver” instead of “Hepatocyte” resulted in 18463 datasets. I then sampled 1k random RNAseq datasets from each set and plotted the results for the 11 hepatocyte marker genes and the 12 Endothelial marker genes (for reference) below (Fig. 1).

Fig 1: ARCHS4 expression for Table S11 genes. Upper plot: hepatocyte marker genes. Lower plot: Endothelial marker genes.

Indeed, expression of Endothelial marker genes is considerably lower, and this possibly explains why less ‘labelled’ genes were found for this gene set.

Is there evidence for whole mRNA (fragments) in the kidney data?

One maybe more convincing analysis would be to look for reads mapping to 3’-ends of genes that are not expressed in the target cell type (kidney) at all but are abundant in the originating one (liver). To demonstrate, I conducted a quick-and-dirty search for genes that are highly expressed in hepatocyte datasets (median TPM > 5k) and lowly in kidney data (median TPM < 10) and plotted the data for these 28 genes (notably, only Mup3 is shared with the CellMarker list) in Fig. 2. When looking at the number of T-to-C containing reads in the labelled kidney datasets, however, I found only marginal numbers (max 3) converted reads in the labelled kidney datasets. Thus, I found no evidence that one of those transcripts was transported from liver (which of course does not exclude that others might have).

Fig 2: ARCHS4 expression for genes that are highly expressed in Hepatocyte datasets and lowly in kidney datasets.

I am also sceptical wrt. the discussion of the authors for distinguishing transferred transcripts (fragments) and dissociated nucleotides (p7ff.). The authors mention 478 mRNAs that were labelled by 4TU in the kidney under both basal and injury conditions, but not by 4sU (Table S11). When looking at their ARCHS4 expression, however, I found several that are very lowly expressed in hepatocyte/liver datasets as shown in Fig. 3, which gives me low confidence that the respective reads stem from hepatocyte-expressed RNAs.

Fig 3: ARCHS4 expression of selected genes that were reported to be labelled by 4TU in the kidney under both basal and injury conditions, but not by 4sU (Table S11). Here, I show 28 genes with very low expression in liver which makes it unlikely that the respective kidney reads stem from liver transcripts.

Altogether, I don't see a robust way for distinguish between reads stemming from kidney and liver in this dataset besides studying the binomial probabilities for T-to-C conversions as discussed above. This binomial probability should correspond to the effective 4sU concentration in the nucleus at transcription time and should arguably be much lower in the target (kidney) cells. So, if intact labelled RNAs are imported from liver into kidney, then they would have been transcribed with much higher 4sU concentrations in place and the 'density' of T-to-C conversions would be higher than for the respective RNAs transcribed in the kidney cells themselves. I doubt, however, that such an analysis (i.e., separating those two sets of RNAs or at least showing that there is more than one such set) is possible given the very low conversion rates in the kidney data and the complexity of such an analysis.

Determining genes with significantly increased T-to-C conversions.

A central claim of the authors is that they detected relatively high percentages (5% under healthy conditions and 34% after acute hepatocellular injury) of, what they call, 'labelled' transcripts. To test for this, if I understood properly, the authors applied a two-group FDR-controlled beta-binomial test to the conversion rate output from SLAMDUNK (and from their own smallRNA pipeline discussed below) to identify genes that are "labelled" in their terminology. I have several questions in this regard:

- As far as I know, the beta-binomial test was proposed for comparing count data, so the authors should explain why this test is appropriate for their purposes and extend the respective description of how the test was conducted in the methods section. Also, the respective data tables including the output from the countdata package (p-values, and adjusted p-values as well as normalized and fold-change values) as well as the abundance (in TPM) per gene should be provided (in the current version, I just found lists of genes that passed the test).
- Additionally, the authors should confirm that this method does not report false positives when applying to sets of genes that are unlabelled in liver data (see marker gene discussion above) or by using *in-silico* simulation data with given (fixed, low) conversion rates/sequencing error. Did the authors apply this approach to subsets of Cre-neg replicates or compare Cre-neg and 4TU-neg data?
- Also, in the text it is mentioned that (in the initial experiment) 0.6% of all analysed genes in the (labelled) kidney QuantSeq dataset were considered as labelled. What was the abundance of those genes? Did the probability to be labelled correlate with a gene's abundance? What fraction of reads were labelled and how does this compare to the liver dataset? How did the dataset with the high conversion rate (cf. Fig2B) influence this number?

Technical description of the Quantseq analysis.

The description of the Quantseq data analysis is insufficient in my opinion and needs to be extended:

- What 3'-end annotations were used and how were they calculated?
- What version of SLAMDUNK was used? What commandline was used?
- The read mismatch rates per read position should be plotted (e.g., SLAMdunk output) and provided for all datasets.
- Did the authors observe increased mismatch rates at the read ends and consider clipping the read 5'-ends? If yes, how many bases were clipped?
- What var_fraction (given the low overall conversion probabilities) was chosen to exclude SNPs?
- SLAMDUNK provides many useful QC statistics (e.g., conversion rate per read or T>C conversions over UTR positions) that can be used to determine biases in the data. Those should be added to the supplement.
- Did the authors consider controlling duplication rates with UMIs?

Technical description of the smallRNA analysis.

The smallRNA results of this study are also of concern. The high abundance of MicroRNAs in exRNA (as reported by many studies), for example, and their increased stability when compared to mRNAs should in principle make it easier to detect them if they are indeed transported from liver to kidney cells. The authors explain not finding labelled smallRNAs in kidney data by "insufficient sensitivity" (not sure what this means) of their approach, but I don't think they have presented enough evidence to exclude technical reasons related to their analysis pipeline. I have several questions/comments in this regard:

- How was the proposed pipeline evaluated? Was synthetic (simulated) data used to validate its proper function?
- Does the pipeline consider 3'/5'-isoforms and untemplated additions (tailing) of small RNAs?
- What duplication rates were observed by the authors? Why were no UMIs used for deduplication?
- How were pre-processed reads mapped to the respective reference sequences? (bowtie2 command line)

Minor comments

- Fig 1B: add box and jitter-plots (to avoid overplotting) and test for significance.
- Page 5: reference “supplemental Tables S4 & S5” should be “supplemental Tables S5 & S6”? All table/figure references should be checked.
- Fig 2G: what are ‘unmapped’ small RNAs? How was the T-to-C conversion rate of those estimated?
- Fig 1: subfigure labels are wrong in the caption.
- Fig 4E: Besides clustering of the “Labelled in injury” data, the PCA also shows clear clustering of the control datasets along PC2. The authors should discuss and investigate (can these differences partly be explained due to transcriptional changes induced by the AAV8-TBG vector?).
- I am missing detailed mapping statistics for the smallRNA datasets of the main experiment (cf. Fig 2I for the initial experiment)
- Software, database and annotation versions & commandlines should be added to the Supplement.
- Table S3: how were conversion rates calculated?
- Table S12: Versions of the annotation datasets are missing.
- The sample sheet I was given during the review process should be published alongside the study data.
- Where can I see T-to-C conversion rates per dataset for the main experiment?

Appendix: Report on my small re-analysis of parts of the Quantseq data.

I have downloaded the datasets from ENA and, thanks to the sample sheet provided by the authors, did a quick and dirty analysis of all Quantseq/labelled and unlabelled datasets:

- I used the SLAMDUNK nextflow pipeline (<https://nf-co.re/slamseq/1.0.0/>)
- I set the following parameters:

```
nextflow run nf-core/slamseq --fasta ${REF} --bed ${BED} --  
trim5 5 --var_fraction 0.1 --read_length 100 --skip_deseq2  
--input slamdunk_sample_sheet.tsv -profile singularity
```
- My reference sequence was GRCm38 (mm10)
- My BED file with 3'-end annotations (SLAMDUNK counting windows) was created from gencode.vM21 annotations. I calculated the (spliced) 3'-end intervals of length 1000bp for each annotated transcript, grouped by gene and flattened the intervals (i.e., merging overlapping ones).
- I trimmed 5 bases from the 5'-end as I observed reduced sequencing qualities in some datasets in a pre-experiment.
- I set the **var_fraction** parameter to 0.1 due to the low conversion rates in this dataset.

Looking at the multiqc summary, SLAMDUNK reported no converted UTRs for the kidney samples and very low conversion rates per read, well within the range of sequencing error (compare, e.g., T>C with C>T and A>G with G>A in for the pink/red samples), Fig 4.

Conversion rates per UTR

This plot shows the individual conversion rates for all UTRs (see the slamdunk docs).

Fig 4: SLAMDUNK QC plots. Top row: Conversion rates per UTR and T>C conversions over read positions. Bottom row: Conversion rates per read for plus and minus strand reads. Colors: dark-blue: liver/labelled, cyan: liver/labelled_APAP, red: kidney/labelled, pink: kidney/labelled_APAP)

When I added all datasets from the 'unlabelled' category, I observed high mismatch rates in the unlabelled/liver but not the unlabelled/kidney data, Fig 5.

Fig 5: conversion rates in reads in same datasets as above, but additionally showing unlabelled/liver (red) and unlabelled/kidney (green) data. Note the apparent differences in mismatch rates despite T>C.